# Teaching Physical Awareness to LLMs through Sounds

**Weiguo Wang** [1]   **Andy Nie** [1 2]   **Wenrui Zhou** [1]   **Yi Kai** [1]   **Chengchen Hu** [1]

## Abstract

Large Language Models (LLMs) have shown remarkable capabilities in text and multimodal processing, yet they fundamentally lack physical awareness–understanding of real-world physical phenomena. In this work, we present ACORN, a framework that teaches LLMs physical awareness through sound, focusing on fundamental physical phenomena like the Doppler effect, multipath effect, and spatial relationships. To overcome data scarcity, ACORN introduce a physics-based simulator combining real-world sound sources with controlled physical channels to generate diverse training data. Using this simulator, we build AQA-PHY, a comprehensive Audio Question-Answer dataset, and propose an audio encoder that processes both magnitude and phase information. By connecting our audio encoder to state-of-the-art LLMs, we demonstrate reasonable results in both simulated and real-world tasks, such as line-of-sight detection, Doppler effect estimation, and Direction-of-Arrival estimation, paving the way for enabling LLMs to understand physical world.

## 1. Introduction

Large Language Models (LLMs) have demonstrated remarkable capabilities in processing and generating text and have successfully expanded into multimodal understanding. However, a fundamental limitation remains: LLMs lack physical awareness–the ability to understand and meaningfully interact with the physical world.

Physical awareness, particularly through sound, is a cornerstone of how humans understand the environment. Sound provides rich, instinctive insights into our surroundings: The Doppler effect, with its frequency shifts, tells us whether a vehicle is approaching or receding; The multipath effect, where sound reflects off surfaces, reveals whether we are in an enclosed space or outdoors; and binaural hearing allows us to pinpoint the sound direction. Apparently, these acoustic cues provide crucial information about physical dynamics and spatial relationships.

Despite their ability to recognize and generate speech, current LLMs can not comprehend the physical properties of sound. This gap poses significant challenges in real-world applications. For example, a voice-controlled vehicle might accept a command such as `Open the Window` from someone outside the car, introducing a security risk due to its inability to discern the physical origin of the voice. Also, in human-robot interaction, embodied AI systems without physical awareness fail to imitate human behaviors, such as turning towards the speaker when addressed.

In this paper, we take an initial step toward bridging this gap by proposing ACORN (Acoustic Channel-Oriented Reasoning Network), a framework that teaches LLMs to develop physical awareness through sound. ACORN equips LLMs with both passive and active acoustic sensing capabilities: Through passive sensing, LLMs can detect line-of-sight (LOS) signal paths, interpret motion via Doppler effects, localize sound sources, and analyze multipath effects. With active sensing, we extend their capabilities to include distance measurement by generating pulses and analyzing echoes, akin to sonar systems. We demonstrate that LLMs empowered by ACORN can not only understand the real world via sound but also interact with it.

However, a significant challenge arises in this endeavor: How to collect and annotate a large-scale dataset that captures diverse physical phenomena? (1) Data collection requires extensive deployment of recording devices across various environments and conditions, making it prohibitively expensive and impractical. (2) Data annotation is nearly infeasible because, unlike text or images where humans can directly annotate content, audio physical phenomena (like Doppler effects or multipath reflections) can not be labeled by human alone and often requires sophisticated measurement equipment.

Our key insight to tackle this challenge is that: *The sound that we hear or microphones capture can be decomposed into two independent components—the sound source and the physical channel through which it travels.* This decomposition suggests an elegant solution: instead of collecting

---

[1]NIO [2]Peking University. Correspondence to: Weiguo Wang <wangwg.wwg@gmail.com>.

*Proceedings of the 42$^{nd}$ International Conference on Machine Learning*, Vancouver, Canada. PMLR 267, 2025. Copyright 2025 by the author(s).

real-world recordings, we can synthesize realistic audio data by combining sound sources from existing datasets with simulated physical channel.

To this end, we devise a dedicate physical channel simulator based on advanced signal processing techniques. This simulator offers complete transparency over the physical channel parameters. It enables precise control over task-specific parameters. Meanwhile, it allows for the systematic randomization of non-critical parameters, ensuring a diverse range of physical channels. By convolving these simulated channels with existing sound sources, this approach facilitates the building of large-scale, diverse sound datasets with annotated physical phenomena.

Leveraging the annotations of the simulated sounds, we curate Question-Answer (QA) pairs that link language with the underlying physical phenomena. To ensure effective alignment between language and audio, we propose a workflow for generating both closed-form and open-form questions and answers that probe the model's understanding of these physical phenomena. In this way, we create AQA-PHY, an Audio QA dataset consisting of 1 million <Audio, Question, Answer> tuples, designed for supervised fine-tuning of LLMs to endow them with physical awareness.

To effectively capture physical information from sounds, we also introduce an effective audio encoder architecture. Unlike traditional audio encoders that primarily focus on content recognition based on magnitude information, our design explicitly incorporates both the magnitude and phase components of sounds. This distinction is crucial, as physical effects often manifest in subtle phase relationships that magnitude-only encoders may overlook (Bai et al., 2020).

We connect our audio encoder to two state-of-the-art LLMs: LLama3.1-8B and Qwen2-7B. We train the model using AQA-PHY. The model achieves strong results across all tasks: 0.924 accuracy in line-of-sight detection, 0.181 MAE in Doppler effect estimation, 0.907 MAE in direction of arrival estimation, 0.903 accuracy in multipath analysis, and 1.599 relative error percentage in range estimation. These demonstrate the feasibility of teaching LLMs to understand physical phenomena through sound.

In summary, our contributions include:

- A physics-based channel simulator enabling generation of diverse audios with various physical phenomena.

- A novel audio encoder capturing both magnitude and phase information for improved physical understanding.

- The AQA-PHY dataset containing 1 million <Audio, Question, Answer> tuples for training LLMs.

- Extensive evaluation demonstrating state-of-the-art performance in both simulated and real-world environments.

## 2. Primer: Sound and Space

The received sound is determined by both its source and the environment through which it travels. When we hear a sound—whether through our ears or microphones—we are receiving a signal that has been transformed by its propagating space. This transformation can be formally modeled as a physical channel filter (Channel Impulse Response, CIR), $h$. Given a sound source $s$, the received signal $y$ is given by

$$y = h \circledast s, \tag{1}$$

where $\circledast$ denotes the convolution operation.

The human auditory system possesses a remarkable capability: from the received signal $y$, it extracts rich spatial information embedded within the channel $h$. This channel contains valuable environmental data, capturing how sound waves interact with surfaces and objects in the environment. Through such acoustic information, humans naturally develop an understanding of their physical surroundings (Traer & McDermott, 2016).

Drawing inspiration from this innate human ability, we aim to endow LLMs with similar auditory capabilities, enabling them to understand physical world through sound.

## 3. Model Architecture

### 3.1. Overview

Figure 1(a) illustrates the model architecture of ACORN. Following established practices in the field, we adopt a common end-to-end architecture (Deshmukh et al., 2023; Gong et al., 2023; Chu et al., 2023; 2024) to ensure the generality of our approach. The architecture consists of three main components: (1) an audio encoder, (2) a text encoder, and (3) an LLM. The audio and text encoders transform raw audio and text inputs into sequences of tokens, respectively. These tokens are then concatenated into a unified sequence. Following common practice, we wrap audio tokens with special tokens: <soa> (start of audio) and <eoa> (end of audio) to establish clear boundaries for the audio content.

The concatenated token sequence is then fed into the LLM, which generates a textual response based on both the text and the audio. As illustrated in Figure 1(a), consider a scenario where a voice command, `Open the Window`, is issued from outside a locked vehicle. Through analysis of the raw audio, the model should be able to detect the absence of a LOS path, indicating that the command originates from outside the secured environment of the vehicle. In such cases, the model's physical awareness enables it to reject potentially unsafe commands.

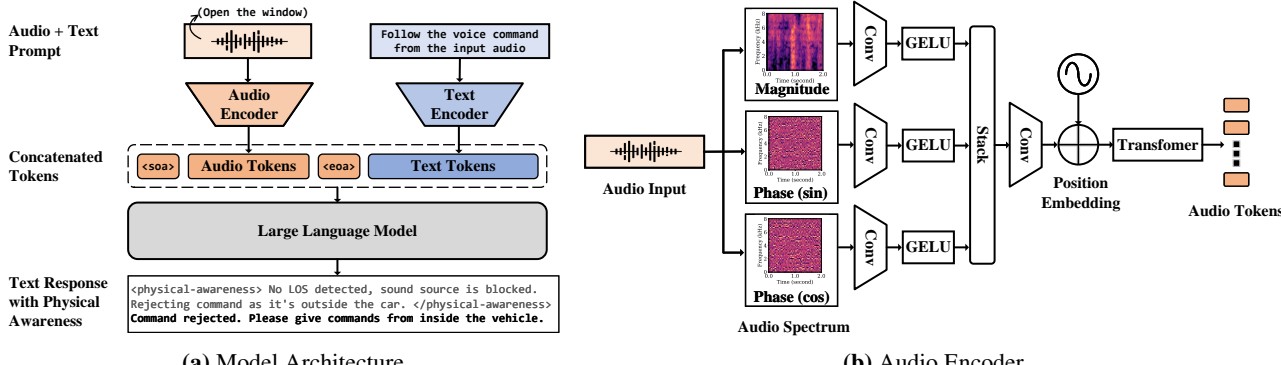

**(a)** Model Architecture        **(b)** Audio Encoder

*Figure 1.* Model architecture and audio encoder design . (a) Model architecture: Integrates audio and text encoders to generate unified token sequences for LLMs, enabling physical reasoning (e.g., detecting LOS). (b) Audio encoder: Processes magnitude and phase components of audio to capture physical phenomena such as Doppler effects and multipath.

## 3.2. Audio Encoder

To effectively understand complex physical phenomena—such as multipath and Doppler effects, or determining time differences between audio channels—an audio encoder must capture fine-grained features beyond sound sources.

Figure 1(b) illustrates our proposed audio encoder. The key design feature is its ability to incorporate phase information through separate channels for the sine and cosine components, in addition to capturing the magnitude of the audio spectrum. By including phase information, our encoder can more effectively capture the characteristics of both the sound source and the acoustic channel, which are essential for understanding the physical world (Bai et al., 2020).

The process first transforms into its spectral components using Short-Time Fourier Transform (STFT). We extract three features from spectral components:

- **Magnitude**: representing the intensity of various frequency components, the magnitude $M(f, t)$ is given by

$$M(f, t) = |X(f, t)|, \tag{2}$$

where $X(f, t)$ is the STFT out.

- **Phase (sin)** and **Phase (cos)**: Instead of using the phase angle directly, we compute the sine and cosine of it. This approach is chosen to mitigate numerical instability caused by phase wraparound, occurring when the phase angle transitions from $\pi$ to $-\pi$ and vice versa. That is

$$\begin{cases} \sin(\theta(f, t)) = \sin(\angle X(f, t)) \\ \cos(\theta(f, t)) = \cos(\angle X(f, t)) \end{cases}. \tag{3}$$

These help in understanding how the sound wave's phase changes, critical for analysis of acoustic channel.

Each of the three components is processed through a 3x3 1D convolution, expanding from 128 to 1280 channels, followed by a GELU activation (Hendrycks & Gimpel, 2016).

The features are then concatenated (3840 channels) and fused through two 3x3 convolutions, reducing the dimensionality to 1280 channels. Sinusoidal positional embeddings are added to retain temporal context, the outputs are processed by a 32-layer transformer, producing the final audio tokens.

## 3.3. Language Model and Training

Here, the language model is responsible for understanding and reasoning about the audio with physical awareness, conditioned by the text prompt. We fuse our audio encoder to two different LLMs LLama3-8B and Qwen2-7B, respectively. The audio tokens are projected to the word embedding size of these models via a linear projection layer.

Both the audio encoder and the LLM are trained jointly on a dataset consisting of <Audio, Question, Answer> tuples. The training objective is to maximize the likelihood of predicting the next token in the Answer, conditioned on both the Audio and Question inputs. Our audio encoder is initialized from Whisper-large-v2 (Radford et al., 2023) to leverage pretrained magnitude representations, while the phase-related subnetwork is trained from scratch to capture fine-grained physical cues critical for physical awareness. LLMs are fine-tuned using LoRA (Hu et al., 2022) to reduce the training workload and to leverage its linguistic capabilities.

## 4. Channel Simulator

Training our model requires a large-scale dataset with annotations of physical phenomena. However, comprehensive data collection in the physical world presents fundamental scalability challenges, given the need to reproduce diverse acoustic phenomena under controlled conditions. To tackle this, we leverage a key insight: *the received signal $y$ can be decomposed into the channel $h$ and the sound source $s$ (see Eq. 1).* This reveals that channel $h$ independently captures

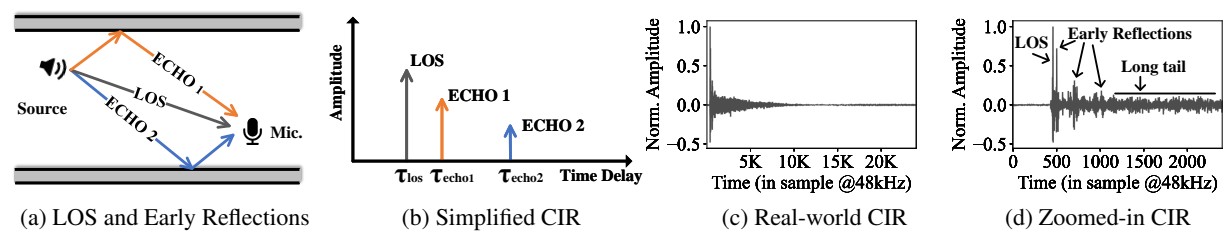

(a) LOS and Early Reflections    (b) Simplified CIR    (c) Real-world CIR    (d) Zoomed-in CIR

*Figure 2.* Illustration of Acoustic Channel.

the physical phenomena, isolating them from source $s$.

Our channel simulator models five main components: (1) LOS paths and (2) early reflections, (3) reverberation, (4) Doppler effects, and (5) microphone array reception. Each can be independently controlled to generate diverse physical phenomena. For more details, please refer to Appendix B.

### 4.1. LOS Path and Early Reflections

We begin with LOS paths and early reflections as they form the basic building blocks of sound propagation.

A CIR models how a sound source propagates and interacts with its environment before reaching the microphone. As depicted in Figure 2(a), these interactions include a direct path (LOS) and two reflected paths, ECHO 1 and ECHO 2.

Figure 2(b) illustrates the CIR function, which can be expressed as a series of delta functions for these paths:

$$h(\tau) = A_1\delta(\tau-\tau_{los})+A_2\delta(\tau-\tau_{echo_1})+A_3\delta(\tau-\tau_{echo_2}),$$
(4)

where $\tau_{los}$, $\tau_{echo1}$, and $\tau_{echo2}$ denote the time delays of the LOS, ECHO 1, and ECHO 2 paths, respectively, and $A_i$ is the corresponding path attenuation.

### 4.2. Reverberation

Reverberation describes the continued interaction of sounds with the environment, creating a prolonged echo effect.

Figure 2(c) presents a real-world CIR, illustrating a sharp initial spike (representing the LOS) followed by smaller spikes (early reflections) and a long diffuse tail (reverberation). Figure 2(d) provides a closer view, offering a more detailed depiction of these components. To effectively capture reverberation, we extend the channel model as follows:

$$h(\tau) = \sum_{i=0}^{N} \alpha_i\delta(\tau - \tau_i) + R(\tau),$$
(5)

where the first sum represents the impulses of LOS and early reflections, and $R(t)$ represents the reverberation tail.

We simulate reverberation $R(\tau)$ by decomposing sound into frequency-specific subbands with unique decay characteristics. We generate Gaussian noise, apply bandpass filters for specific frequency ranges, and modulate with amplitude

envelopes to simulate decay. These subbands are recombined to create a broadband impulse response. Detailed methodologies are provided in Appendix Section B.2.

By controlling the decay rate of $R(\tau)$ and selectively including path components, we can simulate environments ranging from 'dry' spaces with minimal reverberation to those with severe echoes.

### 4.3. Doppler Effect

The Doppler effect, a frequency shift resulting from relative motion between the transmitter and receiver, can be modeled by introducing a time-varying delay in the CIR.

Mathematically, for a transmitter and receiver with a relative velocity $v$, the distance $d(t)$ between them varies as $d(t) = (d_0 + v \cdot t)$, where $d_0$ is the initial distance and $c$ is the sound speed. The time-varying CIR with this dynamic delay can be modeled as:

$$h(t,\tau) = \delta(t - \tau(t)) = \delta(\tau - \frac{d_0 + v \cdot t}{c})$$

We further detail the simulation of the Doppler effect in Appendix Section B.3, along with efficient methods for processing this time-varying CIR.

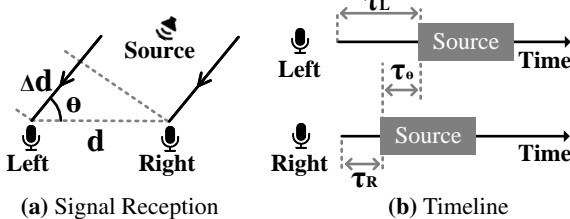

(a) Signal Reception      (b) Timeline

*Figure 3.* Microphone Array Model.

### 4.4. Microphone Array

Sound Direction, formally known as Direction of Arrival (DoA), can be estimated by measuring the Time Difference of Arrival (TDoA) between microphones. Due to their spatial separation, each microphone receives sound waves at slightly different times, enabling DoA estimation.

Figure 3(a) illustrates a binaural recording setup with two microphones placed a distance $d$ apart. When a far-field source (Wang et al., 2020; Benesty, 2008) emits sound from

*Table 1.* Channel Configuration Settings for Different Tasks.

| Task | LOS Path | Early Reflect. | Reverb. | Doppler Effect | Mic. Array |
|------|----------|----------------|---------|----------------|------------|
| LOS Detection | **T** | **R** | **R** | **R** | ✗ |
| Doppler Estimation | **R** | **R** | **R** | **T** | ✗ |
| DoA Estimation | ✓ | **R** | **R** | **R** | **T** |
| Multipath Analysis | **R** | **R** | **T** | **R** | ✗ |
| Range Estimation | **T** | **R** | **R** | **R** | ✗ |

Note: ✓: enabled; ✗: disabled; **R**: randomly enabled/disabled; **T**: controlled target parameter.

direction $\theta$, it creates a differential path length $_\triangle d$. The corresponding TDoA, $\tau_\theta$, is given by

$$\tau_\theta = \frac{_\triangle d}{c} = \frac{d\cos(\theta)}{c} \tag{6}$$

Figure 3(b) visualizes the timeline of sound reception. For these two microphones, let $\tau_L$ and $\tau_R$ denote the delays to reach the left and right microphones. The CIRs, assuming no attenuation, can be simplified as

$$h_L(\tau) = \delta(t - \tau_L) = \delta(t - \tau_{LOS}) \tag{7}$$
$$h_R(\tau) = \delta(t - \tau_L - \tau_\theta) = \delta(t - \tau_{LOS} - \tau_\theta). \tag{8}$$

This general form allows deriving $h_R(\tau)$ by accounting for direction $\theta$, facilitating extension from a single CIR to multiple acoustic channels.

### 4.5. Configuration

The above individual components—LOS path, early reflections, reverberation, Doppler effects, and microphone array reception—combine to create comprehensive acoustic simulations. This modularity allows us to isolate and study specific phenomena while maintaining physical realism.

Table 1 presents the task-specific configurations, with detailed task descriptions provided in Appendix A. For different tasks, we configure simulation components as either enabled (✓), randomly enabled/disabled (**R**), or disabled (✗). When marked as '**T**', the component is systematically controlled for target parameter variation. To maximize channel diversity, we aim to randomly toggle each component whenever possible. However, some tasks have specific requirements—for example, DoA estimation requires the LOS path to be enabled as it relies on direct sound propagation for DoA calculation (Benesty, 2008).

Our simulation approach follows a core principle: targeted physical parameter control with maximal diversity:

**Task-specific Physical Parameterization.** Generally, for each task, we control specific parameters to generate physical channels. For LOS Detection, we control the presence or absence of LOS path to create positive and negative samples.

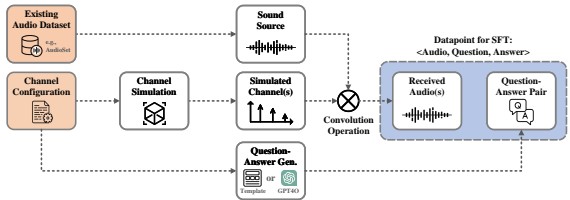

*Figure 4.* Synthesis Workflow of <Audio, Question, Answer>.

For Doppler Estimation, we vary source speeds to generate different frequency shifts. For DoA Estimation, we adjust arrival directions and microphone spacing to create TDoA variations. For Multipath Analysis, we modify decay rates to simulate various reverberation levels. For Range Estimation, we control reflection delays to represent different propagation distances.

**Parameter Randomization.** For each enabled or randomly-enabled component, we randomize internal parameters to enhance channel diversity while maintaining control over task-specific variables. Unlike existing methods that reconstruct CIRs by modeling specific geometric environments (Chen et al., 2022; Zheng et al., 2024), our approach simulates the entire life cycle of sound waves—from emission to reception—by explicitly modeling its key components, including LOS propagation, early reflections, reverberation, and so on. This component-wise, lifecycle-based modeling enables scalable and systematic generation of diverse physical channels, avoiding the complexity and limited generalizability of environment-specific reconstructions.

## 5. The AQA-PHY Dataset

To train our model to understand physical world from sounds, we introduce AQA-PHY, a Audio Question-Answer dataset. Each datapoint is a <**Audio, Question, Answer**> tuple, where audio and question form the input, and answer serves as the target output. Figure 4 illustrates the synthesis workflow of generating a datapoint.

**Sound Source.** The synthesis workflow begins by sampling sound sources from an existing dataset. For this, we utilize AudioSet (Gemmeke et al., 2017), which offers approximately 2 million 10-second sound clips annotated with over 500 labels. This extensive collection provides a rich variety of sound sources, enhancing the dataset's versatility.

**Physical Channel.** For each sound source, we generate a physical channel with specific physical properties using configurations from Section 4.5. The Channel Simulation module transforms these specifications into CIR, which are then convolved with sound sources to calculate a sample of received audio exhibiting desired physical effects.

**Question-Answer Pair.** Two types of QA pairs are de-

*Table 2.* Overall Performance. Values are presented as (Merged | Sole) where "Merged" indicates models trained on combined dataset and "Sole" indicates models trained separately for each task. By default, we focus on Merged results, with Sole results provided for reference.

| Model Architecture | | Task Performances (Merged \| Sole) | | | | |
|---|---|---|---|---|---|---|
| **Audio Encoder** | **LLM** | **LOS Detection** BCA ($\uparrow$) | **Doppler Estimation** $MAE_f$ ($\downarrow$) | **DoA Estimation** $MAE_t$ ($\downarrow$) | **Multipath Analysis** TCA ($\uparrow$) | **Range Estimation** REP ($\downarrow$) |
| Whisper | Llama3.1-8B | 0.867 \| 0.906 | 1.213 \| 3.147 | 5.585 \| 5.601 | 0.845 \| 0.889 | 12.572 \| 17.182 |
| | Qwen2-7B | 0.881 \| 0.910 | 1.042 \| 0.575 | 2.716 \| 6.873 | 0.848 \| 0.897 | 10.609 \| 12.901 |
| ACORN | Llama3.1-8B | 0.920 \| **0.965** | 0.791 \| 0.557 | 1.423 \| 1.349 | 0.890 \| **0.945** | 1.764 \| **1.446** |
| | Qwen2-7B | **0.924** \| 0.962 | **0.181** \| **0.263** | **0.907** \| **1.167** | **0.903** \| 0.944 | **1.599** \| 1.751 |
| Performance on Open QA (Our Encoder + Qwen2-7B) | | 0.898 \| 0.953 | 0.487 \| 0.398 | 2.314 \| 2.043* | 0.906 \| 0.908 | 2.852 \| 1.900* |
| *Random Baseline*** | | 0.50 | 10.00 | 66.67 | 0.33 | 33.33 |

* For both DoA Estimation and Range Estimation in Open QA, LLMs first extract basic measurements from audio signals and then apply step-by-step calculations to determine final results (see example QA in Table 10). For the final calculation results, the MAE of DoA Estimation is 6.505° (Merged) and 5.685° (Sole), while the MAE of Range Estimation is 0.472m (Merged) and 0.352m (Sole).
** Random Baseline values are computed based on task-specific assumptions: random guessing for LOS (BCA = 0.5) and Multipath (TCA = 0.33), random frequency shifts (±15% at 50 m/s) for Doppler, random TDoA within ±100 samples for DoA, and uniform relative errors (0–100%) for Range Estimation.

signed. Examples of QAs are provided in Table 10.

- **Closed-form** pairs focus on classification or quantitative assessments with well-defined answers. These include binary classifications (e.g., "Does the audio contain a LOS path?" with Yes/No answers), multi-class selections (e.g., Rich/Moderate/Negligible for multipath severity), and numerical estimations (e.g., time delays or Doppler shift percentages). These pairs are generated using fixed templates to ensure consistent evaluation.

- **Open-form** pairs elicit detailed explanations of physical phenomena. These questions are phrased in natural language and require comprehensive responses that demonstrate understanding of the underlying physics. For example, in range estimation tasks, answers include step-by-step calculations and explanations of how distance is derived from time-of-flight measurements. We leverage ChatGPT-4 with specific prompts to generate these QA.

**SFT Datapoint.** Finally, the synthesized audio is paired with its corresponding question and answer to form a single datapoint: <Audio, Question, Answer>. This triplet is then formatted into a prompt following the template detailed in Appendix G. For each task, we generate 200,000 closed-form datapoints and 10,000 open-form datapoints.

## 6. Evaluation

### 6.1. Methodology

**Baseline.** We compare two audio encoders: OpenAI's Whisper (Radford et al., 2023) encoder (Whisper-large-v2 with 32 Transformer layers and 0.63 billion parameters), a representative magnitude-only approach widely used in audio processing, and our encoder proposed in ACORN with 0.65 billion parameters. To demonstrate the generalizability of our approach, we pair each encoder with two different large language models (LLMs): Llama3.1-8B-instruct (Dubey et al., 2024) and Qwen2-7B-instruct (Yang et al., 2024). This setting helps verify that our audio encoding improvements are model-agnostic and can benefit various LLM architectures.

**Audio Pre-processing.** The audio pre-processing pipeline transforms raw waveforms (sampled at 16kHz) into spectral representations through Short-Time Fourier Transform (STFT) with a window size of 254[1] and a hop length of 10 ms, chosen specifically to align with the configuration of our baseline, Whisper. Unlike Whisper which converts STFT outputs to mel-spectrograms[2], we directly preserve the full spectral resolution. The process extracts three key components: the log-magnitude spectrum for energy distribution, and sine/cosine components of the phase for temporal information. These features are stacked to create a three-channel representation of the audio.

**Training.** The models are trained using standard next-token prediction loss with the answer text in SFT dataset as labels. We train the models upon MS-SWIFT (Zhao et al., 2024) with modifications to accommodate our specific model ar-

---

[1] Since the input audio is real-valued, the FFT output is symmetric, resulting in (254/2 + 1) = 128 unique frequency bins.

[2] While mel-spectrograms emphasize human speech frequencies, we preserve full spectral resolution for fine-grained analysis of physical world sounds.

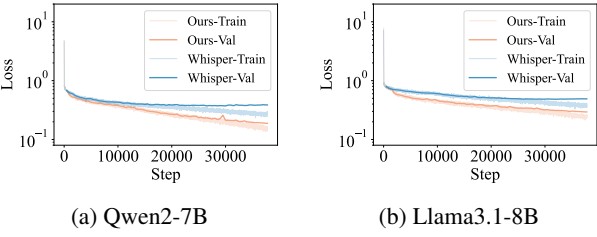

(a) Qwen2-7B          (b) Llama3.1-8B

*Figure 5.* Loss History.

chitecture. The model is trained on 4 NVIDIA A100 GPUs with batch size 32 and completes after 7 epochs. The total training time is about 61 hours. For response generation, the decoding parameters are set with temperature 1, top-p 1, and top-k 50. Appendix C lists more train hyperparameters.

**Evaluation Metrics.** We employ task-specific metrics to evaluate the model across five tasks. (1) For LOS Detection, we use binary classification accuracy (BCA, ↑) to measure the model's ability to correctly identify the presence or absence of LOS path. (2) For Doppler estimation, we employ Mean Absolute Error (MAE) to quantify the model's precision in estimating frequency shift percentage ($MAE_f$). (3) DoA estimation accuracy is evaluated using MAE in time samples ($MAE_s$), measuring the average TDoA deviation between predicted and actual sound source directions.[3] (4) Multipath Analysis performance is evaluated using Triple-class Classification Accuracy (TCA, ↑) across three categories (Rich, Moderate, and Negligible). (5) Range Estimation employs Relative Error Percentage (REP, ↓) to provide a scale-invariant measure of distance estimation accuracy.

### 6.2. Overall Performance

As shown in Table 2, our experimental results validate four main findings: (1) the feasibility of teaching LLMs to understand physical phenomena through sound, as demonstrated by strong performance across all tasks; (2) the superiority of our audio encoder over existing approaches, showing consistent improvements in all evaluation metrics; (3) the model-agnostic nature of our approach, evidenced by similar performance of different LLM architectures; (4) the effectiveness of natural language communication, demonstrated by maintained reasonable performance on open-form QA.

Note that, in Table 2, the evaluation metrics show model performance in the format "(Merged | Sole)", where "Merged" represents models trained on combined datasets for all tasks, while "Sole" indicates models trained separately for each individual task. Our evaluation primarily focuses on the Merged results as the default metric, while Sole results is provided for just reference and comparison.

**Feasibility of Understanding Physical World.** Our exper-

imental results demonstrate the fundamental feasibility of teaching LLMs physical awareness through sound. Across all tasks, our models achieve promising performance - with BCA above 0.92 for LOS detection, $MAE_f$ below 0.8 for Doppler estimation, $MAE_t$ around 1.4 for DoA estimation, TCA above 0.89 for multipath analysis, and REP below 1.8 for range estimation. These strong results across diverse physical tasks demonstrate that LLMs can effectively develop physical awareness through acoustic signals.

**Audio Encoder Superiority.** The audio encoder proposed in ACORN shows clear superiority over the existing Whisper baseline across all evaluation metrics. For LOS detection, we achieve more than 5 percentage points of improvement in BCA (0.920 vs 0.867 with Llama). In Doppler Effect estimation, our $MAE_f$ shows significant reduction (0.181 vs 1.042 with Qwen2). For DoA Estimation, we reduce 74.5% $MAE_t$ (1.423 vs 5.585 with Llama). Perhaps most notably, in Range Estimation, our approach reduces the REP by 7x (1.764 vs 12.572 with Llama), demonstrating substantially more accurate distance estimation capabilities.

The convergence behavior further validates this superiority– as shown in Fig. 5, our approach achieves faster convergence and lower final loss values during training across both Llama and Qwen architectures. This consistent advantage in both convergence rate and final performance suggests that our audio encoder's design - particularly its incorporation of both magnitude and phase information - provides a more effective foundation for learning physical awareness.

**Model-agnostic Property.** The consistent performance improvements across different LLMs indicate that our audio encoding approach is robust and model-agnostic. Both Llama and Qwen show similar patterns of enhancement when paired with our encoder compared to the Whisper baseline, suggesting our method can generalize across different LLM architectures.

**Open QA Performance.** As shown in Table 2, when testing open-form QA with our encoder and Qwen2-7B, the model demonstrates two key capabilities: natural language communication of physical phenomena and multi-step calculations. For direct estimation tasks like LOS detection (0.898) and multipath analysis (0.906), it provides clear explanations in natural language. For tasks requiring multi-step calculations, such as DoA and range estimation, it extracts measurements and applies physical formulas to derive results (DoA error: 6.505°, Range error: 0.472m).

### 6.3. Ablation Study of Acoustic Factors

We conduct ablation studies to understand how different acoustic factors affect the understanding of physical world. While these factors could potentially impact all tasks, we select representative tasks for illustrative purposes to examine

---

[3] The key parameter in estimating DoA is TDoA (see Eq. 6).

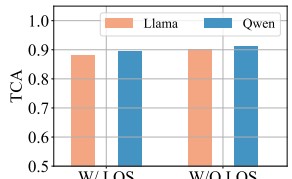
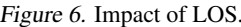

*Figure 6.* Impact of LOS.

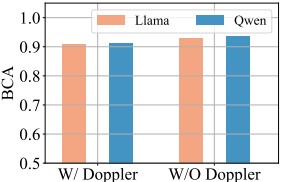
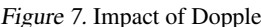

*Figure 7.* Impact of Doppler.

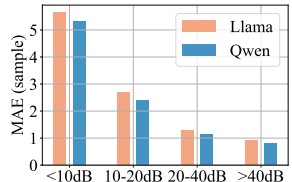

*Figure 8.* Impact of SNR.

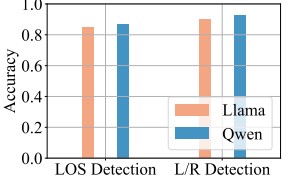

*Figure 9.* Real-World Results.

the effects of various acoustic phenomena. By default, we set the audio encoder to our proposed design.

**Impact of LOS.** Figure 6 demonstrates the impact of LOS paths on multipath analysis by comparing scenarios with and without LOS components. For Llama, TCA decreases from 0.901 to 0.880 when LOS paths are present, while for Qwen, it shows a similar trend with a smaller gap (0.912 to 0.895). This makes intuitive sense, as strong direct-path signals can overshadow the subtle echoes and reflections that characterize reverberation, making it challenging to assess multipath characteristics like reverberation time.

**Impact of Doppler Effect.** Similarly, Figure 7 illustrates how Doppler effects influence LOS detection. The introduction of Doppler shifts slightly reduces the BCA from 0.930 to 0.910 for Llama and from 0.936 to 0.912 for Qwen. This slight decrease in performance is expected, as the Doppler effect can introduce signal distortion, adding complexity to the analysis of LOS components. For a more comprehensive breakdown of the results, see Appendix Section E.

Notably, our method demonstrates strong resilience to these acoustic variations, with slight performance degradations. This robustness makes our approach particularly suitable for real-world environments where multiple acoustic phenomena often occur simultaneously.

**Impact of SNR.** Figure 8 illustrates how signal-to-noise ratio (SNR) affects ranging estimation performance. As expected, estimation accuracy improves significantly with higher SNR levels. For Llama, the MAE decreases steadily from 5.64 samples at low SNR (<10dB) to 0.91 samples at high SNR (>40dB). Qwen shows a similar trend, with MAE reducing from 5.33 samples to 0.80 samples as SNR increases. This trend aligns with the general rule of thumb in acoustic sensing or ranging systems.

### 6.4. Real-World Experiments

**Deployment.** In line with standard practices in vehicle audio systems, four omni-directional microphones are deployed throughout a NIO ES6 vehicle cabin. Two microphones are positioned in the front seat area and two in the back, each with left-right configuration. The audio signals are synchronized and collected via a data acquisition board at 16 kHz sampling rate. Detailed hardware specifications

and data collection setup are provided in Appendix F.

**Results.** Our real-world experiments focused on LOS detection and DoA estimation in a vehicle environment. The model are trained solely on simulated data and evaluated on real-world data in a zero-shot manner. Figure 8 presents the experimental results. For LOS detection, distinguishing between interior and exterior sound sources, the Llama-based model achieved 0.845 accuracy while the Qwen-based model maintained 0.870 accuracy. In the DoA estimation task, implemented as left/right classification within the vehicle cabin, Llama achieved 0.8975 accuracy while Qwen maintained 0.925 accuracy. These real-world accuracies show expected degradation compared to our controlled experimental results, reflecting typical challenges in transitioning from simulated to real-world environments. Despite this performance drop, the results show the practical viability of our approach in the real world.

## 7. Related Work

**Audio LLM.** Recent work has seen significant progress in extending large language models to handle audio modalities. AudioGPT (Huang et al., 2024) and HuggingGPT (Shen et al., 2023) use LLMs as controllers to coordinate multiple specialized audio models for different tasks. In contrast, models like Pengi (Deshmukh et al., 2023), LTU (Gong et al., 2023), and Qwen-Audio (Chu et al., 2023; 2024) take a more unified approach by directly connecting audio encoders with language models to enable end-to-end learning. SpeechGPT (Zhang et al., 2023) introduces a novel direction by discretizing speech into tokens that can be directly processed by LLMs, while BAT (Zheng et al., 2024) and the work (Tang et al., 2024) extend these capabilities to spatial audio understanding.

However, these prior works primarily focus on semantic understanding of audio content or basic spatial properties, without explicitly modeling physical awareness. In contrast, ACORN aims to teach LLMs physical awareness through sound, covering dynamic effects like Doppler, multipath, and range estimation. Furthermore, our approach to audio encoding is distinct from BAT's method. While BAT computes Interaural Phase Difference (IPD) between each pair of microphone channels, our encoder extracts phase infor-

mation directly from individual channels. This approach not only avoids the quadratic growth in pairwise computations as microphone count increases but also preserves complete phase information for each channel, enabling a more general representation of acoustic signal.

**Physical Reasoning in LLM.** Recent works have explored LLMs' capabilities in physical reasoning. NEWTON (Wang et al., 2023) introduces a comprehensive benchmark evaluating LLMs' physical reasoning through text and visual inputs. The work (Ghaffari & Krishnaswamy, 2024) examines multimodal reasoning about physical dynamics in simulated environments. Works such as (Cherian et al., 2024; Memery et al., 2023) investigate whether LLMs can estimate parameters of physical systems. With the rise of generative LLMs, researchers also explore their applications in robotics, where physical understanding is vital. Works such as (Ahn et al., 2022; Liang et al., 2023; Driess et al., 2023) examine LLMs as back-end planners in situated environments. While these efforts primarily focus on physical reasoning through textual and visual modalities, our work explores how acoustic information can empower LLMs to develop physical awareness.

**Acoustic Sensing.** Prior to the emergence of LLMs for audio understanding, acoustic sensing is dominated by handcrafted signal processing techniques such as ToA, TDoA, FMCW, and CIR (Bai et al., 2020; Li et al., 2022). These methods extract physical properties for tasks like activity recognition (Li et al., 2020), gesture tracking (Mao et al., 2016), communication (Wang et al., 2022a), and localization (Wang et al., 2022b;c), but often rely on manual feature design, synchronization, or controlled settings. In contrast, our approach leverages pretrained audio models to learn physical awareness directly from data, potentially offering greater adaptability and scalability.

## 8. Limitations and Future Work

**Multi-turn Conversation.** Our current model is limited to single-turn interactions. Future efforts will aim to incorporate multi-turn conversations, enhancing the ability to guide users through more complex reasoning.

**Internal Thought.** Our approach currently evaluates direct outputs for physical world understanding. Inspired by OpenAI-o1 (OpenAI, 2024) and Deepseek-R1 (Guo et al., 2025), we may examine how LLMs internalize this understanding to enhance response conditioning, thus improving the integration of physical awareness with LLM capabilities.

**Extensive Real-World Experiments.** While our real-world experiments provide valuable insights, they are limited in scope and scale. To fully validate and refine our model's capabilities, more extensive real-world testing is necessary.

## 9. Conclusion

We present ACORN, a framework that teaches LLMs physical awareness through sound. ACORN addresses a fundamental gap in LLMs' ability to understand and interact with the physical world. By introducing a physics-based channel simulator, a novel audio encoder architecture, and the AQA-PHY dataset, we enable LLMs to understand various physical phenomena.

## Acknowledgments

We sincerely thank the anonymous reviewers for their insightful comments and constructive feedback, which helped improve the quality of this work.

## Impact Statement

Our research on teaching physical awareness to LLMs through sound holds significant potential for applications across various fields:

**Empowering Embodied AI.** Equipping LLMs with physical awareness through sound significantly advances embodied AI. By imitating human auditory capabilities, such as turning toward sound sources and adapting to acoustic environments, these systems can engage in more natural and intuitive human-robot interactions. This development enhances usability and user engagement, fostering more effective collaborations between humans and AI.

**Advancements in Safety Applications.** This work addresses a critical need for secure and reliable AI systems by enabling LLMs to reason about physical phenomena such as sound source localization and LOS detection. These capabilities can prevent unauthorized access in safety-critical environments like autonomous vehicles and smart homes by rejecting unsafe voice commands originating from external or unauthorized sources. Furthermore, by integrating acoustic sensing, our approach equips autonomous vehicles with the ability to detect and localize critical sound cues, such as emergency sirens or passenger screams. This advancement addresses a potential limitation in current autonomous driving systems, often described as "deaf drivers".

**Innovations in Channel-Based Simulation** Our channel-based simulation framework can be directly applied to existing audio LLMs, enabling the generation of large-scale datasets that capture diverse acoustic phenomena at significantly lower cost compared to traditional data collection.

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

# A. Task Description

*Table 3.* Tasks for Physical Awareness Enhancement in LLMs

| Task Name | Description | Example Applications |
|---|---|---|
| LOS Detection | Determine if the line-of-sight path is blocked or not. | Reject voice commands from outside a secured area like a car. |
| Doppler Estimation | Estimate the frequency shift in sounds due to the relative motion of the source. | Determine if the source is approaching or moving away. |
| DoA Estimation | Estimate the direction from which a sound originates based on the TDoA. | Enable embodied systems to rotate towards sources, imitating human head movements. |
| Multipath Analysis | Detect the severity of reverberation resulting from multipath effects. | Assess sound quality by identifying distortions due to complex surrounding reflections. |
| Range Estimation | Measure the distance to objects by employing sonar techniques. | Enable proximity sensing to determine phone screen-lock behavior during calls. |

The tasks listed in Table 3 explore both **passive** and **active** approaches to understand physical world. The first four tasks - line-of-sight (LOS) Detection, Doppler (Effect) Estimation, Direction of Arrival (DoA) Estimation, and Multipath (Effect) Analysis - demonstrate passive sensing capabilities, where LLMs analyze incoming sounds to understand physical phenomena. The fifth task, Range Estimation, explores the feasibility of active sensing, where the LLM actively probes the environment by playing, recording, and analyzing sound pulses, similar to how sonar systems operate.

### A.1. Passive Sensing Tasks

Our passive sensing tasks evaluate the ability of LLMs to extract physical information from ambient or environmental sounds:

**LOS Detection** enables LLMs to determine whether there is a direct path exists between the sound source and receiver. When sound travels through a direct path, it arrives earlier than reflections and contains highest energy due to minimal path attenuation. This capability is crucial for security applications, such as validating voice commands in vehicles or buildings. For example, a car's voice control system should reject voice commands from outside the vehicle to prevent unauthorized access. Similarly, smart home devices can use LOS detection to determine if voice commands originate from inside or outside the house.

**Doppler Estimation** focuses on measuring frequency shifts caused by relative motion between sound sources and receivers. When a sound source approaches, the perceived frequency increases due to wave compression; when it recedes, the frequency decreases due to wave expansion. This shift largely depends on the relative velocity. This physical awareness is valuable for various applications, from estimating vehicle speeds to enabling proximity-based responses in smart devices. For instance, a device could adjust its behavior based on whether the sound source is approaching or moving away.

**DoA Estimation** teaches LLMs to determine where sound originates using multiple microphones. This task relies on analyzing time differences of arriving different microphones (i.e., Time Difference of Arrival, TDoA). This task is beneficial to natural human-robot interaction, allowing systems to turn toward heads just as humans do. It might also enable spatial audio processing and directional noise suppression in applications like smart speakers and hearing aids.

**Multipath Analysis** involves understanding how sound reflects and reverberates in different environments. Sound waves bounce off surfaces, creating complex patterns of early reflections and late reverberation. This capability helps systems assess audio quality and adapt to various acoustic environments. For instance, if the LLM detects excessive reverberation degrading sound quality during a voice communication, it could proactively suggest to the other party, 'It seems there's a lot of background reverberations—moving to another place might help.' This level of adaptive response enables the LLM to function similarly to a human listener, enhancing user experience by facilitating clearer communication in less-than-ideal acoustic conditions."

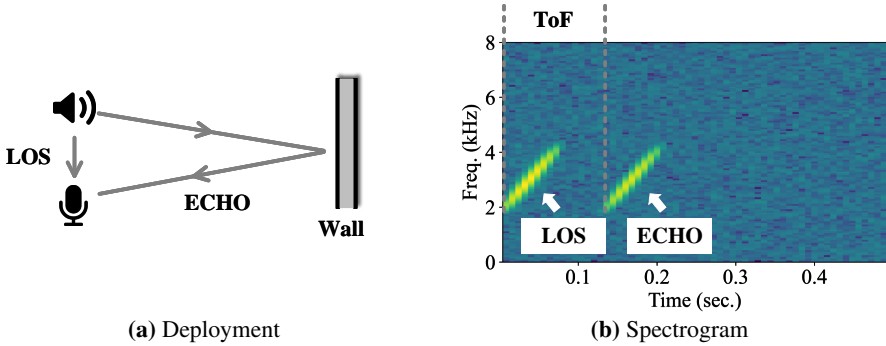

**(a)** Deployment        **(b)** Spectrogram

*Figure 10.* Ranging Estimation Illustration.

## A.2. Active Sensing Tasks

**Range Estimation** explores active sensing capabilities, requiring LLMs to initiate measurements by transmitting sound pulses and analyzing their echoes. Unlike passive tasks that only observe existing sounds, this represents a significant advancement toward active environmental interaction. We envision that the LLMs can trigger pulse transmission through function calls, measure echo return times, and calculate distances based on the speed of sound. This active approach enables precise distance measurements independent of ambient sound conditions, demonstrating how LLMs can actively engage with their physical environment.

To better understand this, we give an example in Figure 10. As shown in Figure 10(a), the basic deployment involves a system that actively transmits a sound pulse. This pulse travels along two paths: a direct Line-of-Sight (LOS) path to the microphone, and a reflected ECHO path that bounces off the wall. This configuration mirrors the principles used in natural and technological sonar systems. The spectrogram in Figure 10(b) illustrates how these paths manifest in the received signal, demonstrating the feasibility of distance estimation. We can clearly observe two distinct bright patterns: the LOS signal arrives first, followed by its ECHO from the wall. The time difference between these signals, marked by the Time-of-Flight (ToF) dashed lines, makes distance calculation straightforward. Since we know sound travels at approximately 343 meters per second in air, and the echo travels twice the distance to the wall (to the wall and back), we can determine the wall distance by taking half of the total distance that the echo traveled.

We want to mention that, range estimation is unique among our tasks not only in its active sensing approach but also in its sound source. While other tasks utilize real-world sound sources sampled from the AudioSet dataset, Range Estimation employs specially modulated signals for transmission. Specifically, it uses Frequency-Modulated Continuous Wave (FMCW) signals, also known as *chirp*, where the frequency changes linearly with time. These chirp signals, widely used in Radar and Sonar systems (Rao, 2017), provide excellent range resolution properties. To ensure the diversity of source sources used for range estimation, we randomize several chirp generation parameters including the starting frequency, ending frequency, and chirp duration.

## B. Implementation Detailed for Channel Simulator

### B.1. LOS ad Early Reflections

In Eq. 4, we use the Delta function ($\delta$) to explain CIR for better understanding. In practice, we employ damped sinusoids, which provide both precise parameter control and naturally model physical acoustic behaviors like attenuation and resonance. Mathematically, Each path in the CIR is modeled as a damped sinusoid

$$z(t) = \cos(2\pi f_z t)e^{-\alpha_z t}, \tag{9}$$

where $f_z$ is the sinusoid wave frequency, $\alpha_z$ is the decay rate, and $t$ is the time. Therefore, Eq. 4 can be rewritten as

$$h(\tau) = A_1 z(\tau - \tau_{los}) + A_2 z(\tau - \tau_{echo_1}) + A_3 z(\tau - \tau_{echo_2}). \tag{10}$$

Table 4 lists the corresponding parameters used to simulate LOS and early reflections. Except amplitude, The parameters for LOS paths are largely fixed to model the direct path's consistent nature, as it experiences minimal environmental interactions. In contrast, the parameters for early reflections vary randomly to capture the diversity of channels.

*Table 4.* Parameter Specifications for Damped Sinusoid Path Generation

| Parameter | Description | Unit | Value/Range | | Sampling Method | |
|---|---|---|---|---|---|---|
| | | | LOS | Early Reflections | LOS | Early Reflections |
| Base Frequency ($f_c$) | Sinusoid wave frequency | Hz | 2,000 | 2,000 | Fixed | Fixed |
| Frequency Modification | Factor for tuning base frequency | - | 1.0 | 0.8 - 1.2 | Fixed | Uniform |
| Decay Rate ($\alpha_z$) | Base exponential decay | Hz | 8,000 | 8,000 | Fixed | Fixed |
| Decay Modification | Factor for tuning decay rate | - | 1.0 | 1.0 - 1.5 | Fixed | Uniform |
| Amplitude ($A$) | Path strength | - | 5.0 - 20.0 | 0.1 - 0.5 | Uniform | Uniform |
| Duration | Time duration of path | second | 0.01 | 0.01 | Fixed | Fixed |

## B.2. Reverberation

During the simulation of reverberation, Gaussian noise is first generated as the base signal, characterized by a normal distribution with a mean of zero and a standard deviation of one, denoted as

$$n(t) \sim \mathcal{N}(0, 1). \tag{11}$$

The noise is generated with a sampling rate of 16 kHz, and to accommodate the requirements of our simulation, the duration of $n(t)$ is set to a maximum of 2 seconds, translating to 32,000 samples. This establishes the temporal extent of the impulse response we aim to simulate.

Subsequent processing involves bandpass filtering this noise to isolate specific frequency bands, crucial for accurately modeling the frequency-selective behavior observed in real-world reverberation, where different frequency bands decay at different rates. Each subband is obtained using a Butterworth bandpass filter, selected for its flat frequency response within the passband and sharp cutoff characteristics. The frequency range for each subband $i$ is specified by $f_i^{low}$ and $f_i^{higi}$. The filtered signal for the subband $i$ is given by

$$s_i(t) = \text{bandpass}(n(t), f_i^{low}, f_i^{high}). \tag{12}$$

Each filtered signal $s_i(t)$ is modulated by an amplitude envelope to simulate the natural decay characteristics of sound in an environment. This decay is typically exponential, mathematically represented as $e_i(t) = \exp(-\lambda_i t)$, where $\lambda_i$ represents the decay rate for subband $i$.

Finally, the processed signals from each subband are summed to construct the broadband reverberation, defined as:

$$R(t) = \sum_{i=1}^{N_{band}} s_i(t) \cdot e_i(t), \tag{13}$$

where $N_{band}$ is the number of subbands.

For a detailed breakdown of the parameters used in our reverberation simulation, including the specific frequency bands and decay rates employed, refer to Table 5.

## B.3. Doppler Effect

This subsection provides an detailed explanation of the method for simulating the Doppler effect within a time-varying CIR and explains an efficient processing approach. Instead of using a traditional convolution operation, we apply the Doppler-induced time-varying delay transformation directly to the transmitted signal, reducing computational complexity.

In a stationary channel, the Channel Impulse Response (CIR), $h(\tau)$, is typically constant over time, representing a fixed delay or set of delays for each multipath component. However, for a moving transmitter or receiver, the CIR must vary with time to capture the Doppler effect, which introduces a continuous frequency shift due to the relative motion.

To model this, we introduce an additional time dependency $t$ into the CIR, such that $h(t, \tau)$ captures how the path delay $tau(t)$ changes with relative speed.

*Table 5.* Parameter Specifications for Simulating Reverberation

| Parameter | Description | Units | Value/Range | Sampling |
|---|---|---|---|---|
| Sampling Rate ($fs$) | Sampling rate of audio signal | Hz | 16,000 | Fixed |
| Duration ($T$) | Total length of the Gaussian noise signal | second | 2 | Fixed |
| Number of Samples | Total number of audio samples | sample | 32,000 | Fixed |
| Number of Subbands ($N_{\text{band}}$) | Number of frequency bands for filtering | - | 6 | Fixed |
| Subband Frequency ($f_i^{low}$, $f_i^{high}$) | Lower and upper frequency bounds of subband $i$ | Hz | (50, 200) (200, 500) (500, 1,000) (1,000, 2,000) (2,000, 4,000) (4,000, 8,000) | Fixed |
| Decay Rates ($\lambda_i$) | Amplitude decays rate for subband $i$ | - | $1.5 - 60^*$ | Uniform |
| Butterworth Filter Order | Order of the Butterworth bandpass filter | - | 4 | Fixed |

$^*$ A larger decay rate $\lambda_i$ means the sound energy decays more quickly, resulting in a shorter reverberation time (RT60). Specifically, a $\lambda_i$ value of 1.5 yields an RT60 of approximately 2 seconds, while a $\lambda_i$ value of 60 shortens the RT60 to about 0.05 seconds.

Let's consider a scenario where the transmitter and receiver are separated by an initial distance $d_0$ and moving at a relative speed $v$. As they move, the distance between them changes linearly with time: $d(t) = d_0 + v \cdot t$. Since the speed of sound is $c$, this changing distance translates to a time-varying delay $\tau(t)$ in the signal:

$$\tau(t) = \frac{d(t)}{c} = \frac{d_0 + v \cdot t}{c}. \tag{14}$$

This delay means that the time it takes for each part of the transmitted signal to reach the receiver continuously varies as the distance changes. We can represent this time-varying delay in the CIR by using a delta function that shifts in time. The CIR with Doppler effect then becomes:

$$h(t, \tau) = \delta\left(\tau - \frac{d_0 + v \cdot t}{c}\right). \tag{15}$$

To obtain the received signal $y(t)$, we convolve the transmitted signal $x(t)$ with the time-varying CIR $h(t, \tau)$:

$$
\begin{aligned}
y(t) &= \int_{-\infty}^{-\infty} x(\tau) h(t, t - \tau) d\tau \\
&= \int_{-\infty}^{-\infty} x(\tau) \delta\left((t - \tau) - \frac{d_0 + v \cdot t}{c}\right) d\tau \\
&= x\left(t - \frac{d_0 + v \cdot t}{c}\right)
\end{aligned}
\tag{16}
$$

This expression can be rewritten as:

$$y(t) = x\left((1 - \frac{v}{c})t - \frac{d_0}{c}\right). \tag{17}$$

As we can see, the factor $(1 - \frac{v}{c})$ compresses or stretches the time variable, This aligns with the Doppler effect, where relative motion changes the observed frequency.

More importantly, This expression shows that the Doppler effect can be viewed as resampling the original source $x(t)$ at a new effective sampling rate of $1 - \frac{v}{c}$. By interpreting the delay transformation as a resampling problem, we can achieve the Doppler simulation efficiently using 1D interpolation to calculate the received signal $y(t)$.

For connivent, Table 6 summarizes the parameters relevant for simulating the Doppler effect.

### B.4. Microphone Array

Similarly, Table 7 summarizes the key parameters for simulating the reception of microphone array.

*Table 6.* Parameter Specifications for Simulating Doppler Effect

| Parameter | Description | Units | Value/Range | Sampling |
|---|---|---|---|---|
| Sampling Rate ($f_s$) | Rate at which the signal is sampled | Hz | 16000 | Fixed |
| Initial Distance ($d_0$) | Starting distance between transmitter and receiver | m | 0.5 – 100 | Uniform |
| Relative Speed ($v$) | Relative velocity between transmitter and receiver | m/s | -50 – 50 | Uniform |
| Sound Speed ($c$) | Speed of sound | m/s | 343 | Fixed |
| 1D Interpolation Method | Interpolation technique for resampling | N/A | Linear | Fixed |

*Table 7.* Parameter Specifications for Simulating the Reception of Microphone Array

| Parameter | Description | Units | Value/Range | Sampling |
|---|---|---|---|---|
| Inter-microphone Distance ($d_{mic}$) | Distance between left and right microphone | cm | 8 – 15 | Uniform |
| Direction of Arrival ($\theta$) | The direction in which the sound source arrive at the microphone array | degree | 0 – 180 | Uniform |

## C. Training Hyperparameters

The main training hyperparameters configured are summarized in Table 8.

*Table 8.* Training Hyperparameters

| Parameter | Value |
|---|---|
| GPUs | 4 NVIDIA A100 |
| Global Batch Size | 32 |
| Epochs | 7 |
| Optimizer | AdamW |
| Optimizer Parameters | $\beta_1 = 0.9$, $\beta_2 = 0.95$, $\epsilon = $ 1e-8 |
| Learning Rate Schedule | WarmupDecayLR |
| Weight Decay | 0.1 |
| Warm-up Min Learning Rate | 0 |
| Warm-up Max Learning Rate | 0.0001 |
| Warm-up Ratio | 0.05 |
| LoRA Rank | 8 |
| LoRA Alpha | 32 |
| LoRA Dropout | 0.05 |

## D. Ablation Study of LoRA Rank

We keep the LoRA rank relatively low, 8, to preserve the LLM's language ability, as most representation learning occurs in the audio encoder. This choice is aligned with the model-agnostic property of our model. Higher ranks showed minimal benefit but increased cost (see Table 9).

## E. Detailed Analysis of Doppler Effect on LOS Detection Performance

Figure 11 demonstrates how varying Doppler speeds affect LOS detection for the models Llama and Qwen. The analysis covers speed bins from -50 to +50 m/s. Both models show consistently stable performance across these speed ranges. Notably, performance variations are evident in conditions of higher speeds, underscoring the models' varied sensitivity to distortions caused by the Doppler effect.

*Table 9.* Ablation Study of LoRA Rank

| Setting | | Task Performances (Merged) | | | | |
| --- | --- | --- | --- | --- | --- | --- |
| Model Architecture | LoRA Rank | LOS Detection BCA (↑) | Doppler Estimation MAE$_f$ (↓) | DoA Estimation MAE$_t$ (↓) | Multipath Analysis TCA (↑) | Range Estimation REP (↓) |
| | 4 | 0.919 | 0.869 | 1.631 | 0.881 | 1.943 |
| Our Encoder + Qwen2-7B | 8 | **0.924** | **0.181** | 0.907 | **0.903** | 1.599 |
| | 16 | 0.908 | 0.193 | 0.840 | 0.881 | **1.561** |
| | 32 | 0.910 | 0.187 | **0.731** | 0.890 | 1.724 |

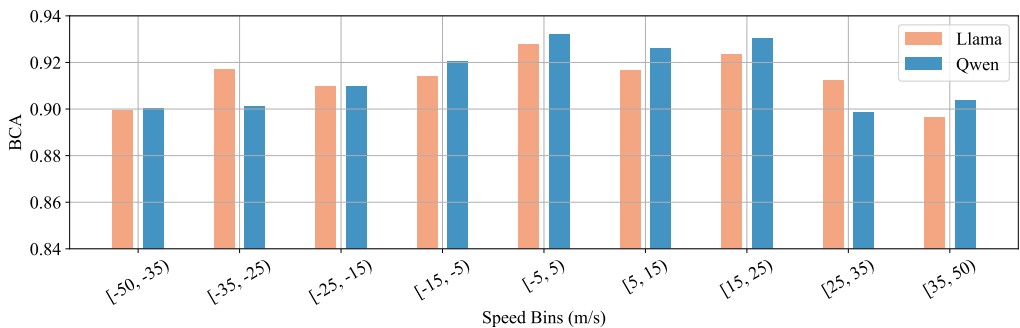

*Figure 11.* Impact of Doppler Speed on LOS detection.

## F. Real-World Deployment

The data collection setup, as illustrated in Figure 12, consists of four omni-directional SPK0641HT4H digital microphones deployed throughout a vehicle cabin (specifically in a NIO ES6, a smart vehicle with Audio Assistant support). Two microphones (Mic. 1 and Mic. 2) are positioned in the front seat area, with one on the left and one on the right, as shown in Figure 12(a). The remaining two microphones are positioned in the back seat area, following the same left-right configuration, depicted in Figure 12(b) and 9(c).

The microphones are driven and sampled by an XMOS XU216 data acquisition board (visible in Figure 12(a)), which ensures synchronization across all four channels. Operating at a sampling rate of 16 kHz, the board streams the synchronized audio signals to a laptop via USB UAC 2.0 protocol for data collection and storage.

For comprehensive evaluation, we place a speaker at various test locations. For exterior testing, we deploy the speaker outside at all four car doors (Figure 12(d) shows an example of one door placement). The speaker was also positioned at various interior locations including the front seat (Figure 12(e)) and back seat (Figure 12(f)).

For data collection, we gather 200 samples for the LOS detection task, evenly split between interior (100 samples) and exterior (100 samples) positions. For the left/right detection task, we collected 400 samples in total, with 100 samples from each of the four seating positions (left-front, right-front, left-back, and right-back seats).

## G. Prompt Format

To ensure the generality and broad applicability, we adopt a standardized prompt format commonly used in the field, similar to those employed by models such as Llama and Qwen. The basic format for single-audio tasks is structured as follows:

```
<|im_start|>system{\n}
You are a helpful assistant.
<|im_end|>{\n}
<|im_start|>user{\n}
Audio 1: <|audio_bos|><|AUDIO|><|audio_eos|>{\n}
{Question}
```

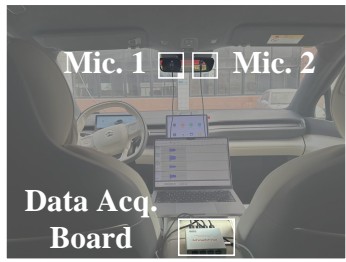

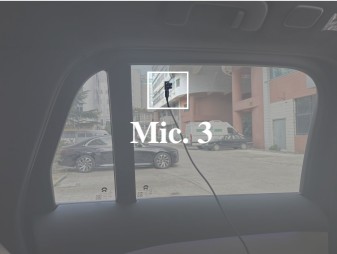

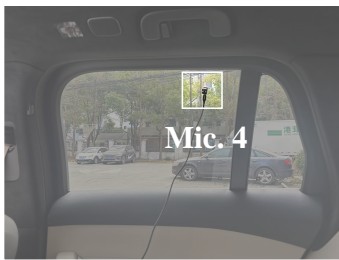

(a) Mic. 1 and 2, and data acq. board

(b) Mic. 3

(c) Mic. 4

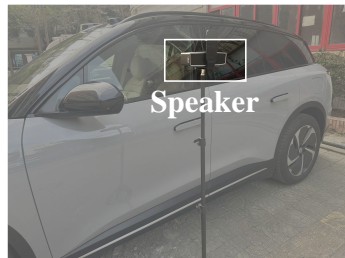

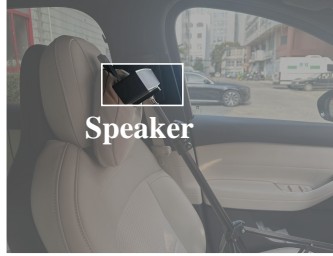

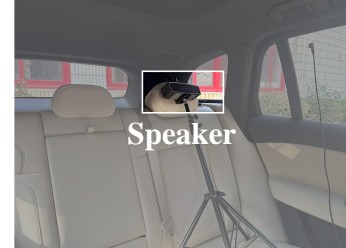

(d) Speaker outside the car

(e) Speaker at front seat

(f) Speaker at back seat

*Figure 12.* Illustration of microphone and speaker deployment in the NIO ES6 vehicle. Microphones (Mic. 1–4) are positioned in the front and back seat areas, while the speaker is placed at various locations for replaying audios.

```
<|im_end|>{\n}
<|im_start|>assistant{\n}
{Answer}
<|im_end|>
```

The format uses "You are a helpful assistant" as the default system prompt. The placeholders {Question} and {Answer} are replaced with the specific question and answer pairs from the dataset. For tasks requiring two audio inputs, we extend the format by adding a second audio segment:

```
<|im_start|>system{\n}
You are a helpful assistant.
<|im_end|>{\n}
<|im_start|>user{\n}
Audio 1: <|audio_bos|><|AUDIO|><|audio_eos|>{\n}
Audio 2: <|audio_bos|><|AUDIO|><|audio_eos|>{\n}
{Question}
<|im_end|>{\n}
<|im_start|>assistant{\n}
{Answer}
<|im_end|>
```

## H. Comparison Between Real and Synthesized CIRs

To qualitatively validate the diversity of our synthesized CIRs, we retrieve the closest match for several real CIRs using similarity search. Representative examples are shown in Figure 13.

## I. Example QA Pairs

Table 10 presents the example QA pairs for five different tasks, including both close and open types.

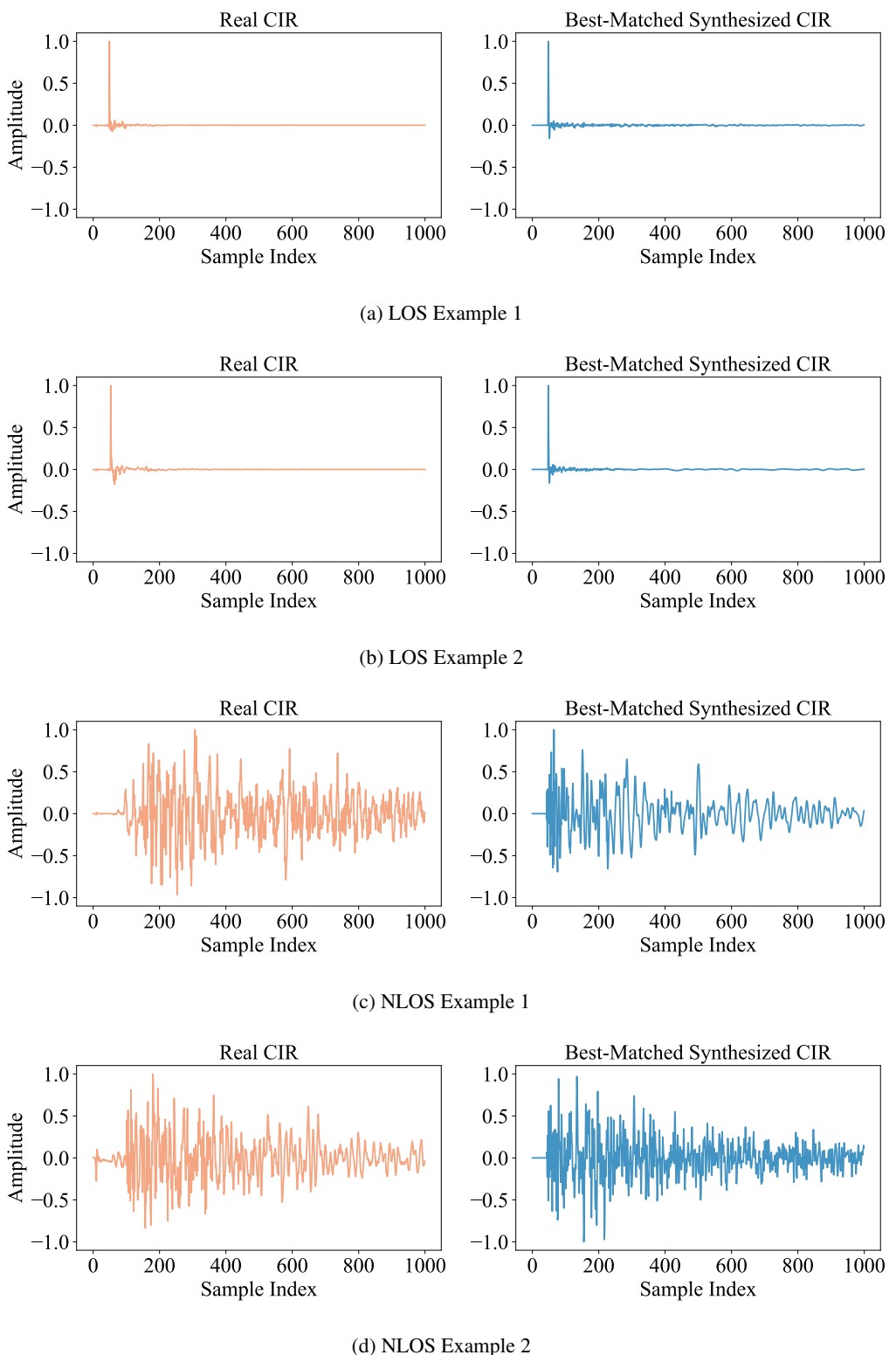

*Figure 13.* Comparison between real and synthesized CIRs in both LOS and NLOS scenarios. For each example, we select a real CIR (left) and retrieve the most similar synthesized CIR (right) using a similarity search. Two LOS and two NLOS cases are shown to demonstrate the diversity and realism of our synthesized CIRs.

*Table 10.* Example QA Pairs for Tasks.

| Task | Form | Example Question | Example Answer |
|---|---|---|---|
| LOS Detection | Close | Audio 1: <Audio> Does the audio contain a line-of-sight path? | {Yes. | No.} |
| | Open | Audio 1: <Audio> Is there a clear LOS path in the audio, or not? | No, the audio doesn't have a clear LOS path. I think the source reaches the receiver through indirect paths. |
| Doppler Effect | Close | Audio 1: <Audio> Audio 2: <Audio> You are provided with a two-channel audio recording. The first channel is the origin the sound. The second channel is the received audio. Please determine whether the sound source is approaching or receding. | {Approaching. | Receding.} |
| | Close | Audio 1: <Audio> Audio 2: <Audio> You are provided with a two-channel audio recording. The first channel is the origin the sound. The second channel is the received audio. Please estimate the Doppler shift percentage. | {XX%.} |
| | Open | Audio 1: <Audio> Audio 2: <Audio> How can the percentage of Doppler shift be estimated from a two-channel audio recording, where the first channel represents the original sound and the second channel captures the received audio? | The estimated Doppler shift is approximately 2.5%. The sound seems to approach you due to a noticeable increase in frequency. |
| DoA Estimation | Close | Audio 1: <Audio> Audio 2: <Audio> Given a two-channel audio recording, where the first channel is from the left microphone and the second from the right, which microphone receives the sound first? | {Left. | Right. | Front.} |
| | Close | Audio 1: <Audio> Audio 2: <Audio> Given a two-channel audio recording, where the first channel is from the left microphone and the second from the right. Please estimate the time difference of arrival. | {XX samples.} |
| | Open | Audio 1: <Audio> Audio 2: <Audio> In a smart home system, two microphones are placed d = 10 cm apart on the ceiling to localize a sound source, such as a speaker, in the living room. Given the recorded audio signals from both microphones and assuming the speed of sound c = 343 m/s with sampling rate fs = 16000 Hz, what is the Direction of Arrival (DoA) of the sound source? | Let's solve this step by step:\n 1. From the audio signals analysis, we estimate the time delay:\n N = -4 samples\n 2. Convert the time delay to time ($\tau$) in milliseconds:\n $\tau$ = N / fs = -4 / 16000 = -0.25 ms\n 3. Apply the DoA formula:\n $\theta$ = arccos(($\pi \times$ c) / d) = arccos((-0.25 / 1000 $\times$ 343) / 0.1)\n \n 4. Simplify the calculation:\n $\theta \approx$ arccos(-0.8575) = 0.5404 radians\n \n 5. Convert radians to degrees:\n $\theta_{deg}$ = ($\theta$ / $\pi$) $\times$ 180 $\approx$ 30.96 degrees\n \n Final Answer:\n The estimated Direction of Arrival (DoA) angle is approximately 30.96 degrees. |
| Multipath Analysis | Close | Audio 1: <Audio> How severe is the multipath effect in this audio? | {Rich. | Moderate. | Negligible.} |
| | Open | Audio 1: <Audio> Could you evaluate the effect of multipath interactions within the given audio piece? | The multipath effect is very rich in this audio, with extensive reverberation creating a complex sound profile. |
| Range Estimation | Close | Audio 1: <Audio> You are given audio data containing both the transmitted pulse and the echo. Please estimate the time of flight. | {XX ms.} |
| | Open | Audio 1: <Audio> You are given acoustic data that includes a transmitted sound pulse and its reflection. Please estimate the distance to the object. | The time of flight for the sound pulse is 59.0 ms. Assuming the speed of sound in air is 343 m/s, the round trip distance to the temperature inversion layer is calculated as 343 multiplied by 59.0 / 1000, equaling 20.2 meters. The altitude of the layer is half of this value, which equals 10.1 meters. |

