# OpenReview forum: "Teaching Physical Awareness to LLMs through Sounds"
_ICML.cc/2025/Conference — ICML 2025 poster_

### Official Review · Reviewer_9yAo · 2025-03-10

**Overall Recommendation:** 4

**Summary:**

This paper proposes a method to incorporate physical awareness into LLMs using audio signals, focusing on fundamental acoustic phenomena such as the Doppler effect, multipath reflections, and direction-of-arrival. The authors introduce a specialized simulator that generates large-scale training data for these phenomena by combining real-world audio clips with artificially controlled channel parameters. They then build an AQA-PHY dataset and design a phase-aware audio encoder to feed signals into two different LLM backbones. Through simulation-based experiments and a limited real-world test in a vehicle cabin, the paper demonstrates that the resulting system can detect line-of-sight, measure Doppler shifts, localize sound sources, analyze reverberation, and even perform active distance measurements.

## Update after rebuttal
After the discussion, I raised my rating since the authors addressed my concerns.

**Claims And Evidence:**

The main claim is that LLMs can be taught physical awareness by training on synthetic and limited real-world audio data. The authors support this claim with consistent accuracy and error metrics across multiple simulated tasks (e.g., line-of-sight detection reaching over 0.92 accuracy, Doppler estimation with low mean absolute error). They also conduct a small real-world test, showing the approach can generalize to actual environments, though on a limited scale. The evidence is mostly numerical, with relevant quantitative metrics across tasks. The real-world demonstration solidifies the claim, but more extensive in-the-wild evaluations might be needed to confirm broader robustness.

**Essential References Not Discussed:**

The current references are sufficient. However, the authors might consider adding more connections to the classic acoustic localization or radar-based distance-estimation work.

**Experimental Designs Or Analyses:**

1. Experiments address five tasks, which I think are comprehensive simulations. However, the real-world experiment is small-scale -- only in a single vehicle scenario. It would be good to see expansions into other real-world environments (e.g., open areas, different types of indoor spaces).
2. Section 6.4 (Real-World Experiments) would benefit from clearer explanations of how well the training data’s synthetic conditions match (or differ from) the actual in-cabin sound environment.
3. The formulas in Appendix B (channel simulator design) are mostly coherent, but I wonder if there could be more detail on how the parameters (e.g., decay factors in reverberation, time-varying delays in Doppler) transfer to real-world conditions.

**Methods And Evaluation Criteria:**

The authors use a physics-based simulator to synthesize channel impulse responses, which are convolved with audio signals to generate corresponding question-answer pairs. Each QA pair is either a closed-form question (e.g., classification or numerical estimation) or an open-ended reasoning question. This approach appears well-structured for the goal of training an LLM to interpret and reason about acoustic phenomena. Their choice of standard metrics aligns with typical acoustic analysis benchmarks.

**Other Comments Or Suggestions:**

A table or figure that directly compares the simulator’s impulse response to measured impulse responses from real rooms or real vehicles would help illustrate simulation fidelity.

**Other Strengths And Weaknesses:**

Addtional Weaknesses:
1. The proposed model is tested on a relatively small set of tasks (five in total). Although these tasks are well-motivated, it would be nice to see an even broader range of physically grounded tasks (e.g., multiple source separation, room geometry estimation, etc.).
2. While the paper claims LLMs can integrate active sensing, the experimental coverage is a bit brief. If distance measurement is a key highlight, it could be valuable to show more step-by-step error analysis in complex real scenarios (moving objects, multiple reflectors).

**Questions For Authors:**

1. Could you provide more details on how the simulator’s parameters (e.g., decay factors for reverberation or velocity for Doppler) were selected or tuned to align with real acoustic measurements?
2. For the active sensing (range estimation), how consistent were the echo results across different real objects and distances? Does clutter (many reflectors) degrade performance significantly?

**Relation To Broader Scientific Literature:**

Integrating audio-based physical reasoning into language models builds on lines of work in multimodal LLM development, such as prior methods that fuse audio encoders with text decoders (e.g., Qwen2-Audio-Instruct, Audio Flamingo, etc.). However, the unique twist here is on physical channel simulation rather than purely content-based audio tasks. This can be applied to some robotics/embodied-AI that gives LLMs actual or synthetic sensor data to reason about physical surroundings. The paper thus extends existing multimodal approaches with a new focus on physical channel cues.

**Theoretical Claims:**

The paper does not delve deeply into new theoretical proofs, but it provides a detailed derivation of how Doppler-shifted signals can be modeled by resampling. The channel modeling steps seem correct and are explained thoroughly. I did not find formal proofs that could be challenged.

---

> ### Author Rebuttal · Authors · 2025-03-30
>
> # Response to Reviewer 9yAo
>
> We sincerely thank the reviewer for the thoughtful and constructive feedback. We are encouraged by your recommendation! Our responses are as follows:
>
> ---
>
> **Regarding Simulator-to-Reality & Parameter Selection**
>
> **Response:** Our simulation approach follows the principle: **maximize parameter randomization while preserving physical realism**. We systematically randomize channel parameters like path number, strength, delay, and reverberation decay to ensure broad coverage of real-world conditions rather than fitting specific environments.
>
> Take in-cabin case as an example.
> - Vehicle interiors typically exhibit 3–5 strong reflections from surfaces like side windows, windshields, dashboards, and ceilings. Our simulator randomly set the number of reflection paths and randomizes their amplitudes and delays to replicate such conditions.
> - For reverberation, real-world RT60 values in vehicles range from 0.2–0.5 s. Our simulator spans a much broader RT60 range of 0.05–2.0s (Table 5), covering cases from 'dry' environments to highly reverberant spaces.
>
> Similarly, we randomize relative velocities from -50 to +50 m/s for Doppler simulation, and directional parameters (e.g., DoA angles from 0–180° and inter-microphone spacing from 8–15 cm) to encompass common real-world sensor setups.
>
> This extensive randomization strategy intentionally produces training conditions more diverse than any individual environment, enhancing robustness and generalization. Our real-world experiments in a vehicle cabin (Section 6.4) confirm that models trained on these diverse simulations transfer effectively, even without environment-specific tuning.
>
> We will clarify these in the revision.
>
>
>
> ---
>
> **Regarding Real-World Evaluation Scope:**
>
> **Response:** We agree that the scope of our real-world evaluation is currently limited, and we sincerely appreciate the reviewer’s suggestion. Our primary goal in including the real-world experiment was to establish a **proof of concept**—demonstrating that models trained purely on synthetic data (AQA-PHY) can generalize effectively to real acoustic environments, thereby validating the feasibility of sim2real transfer.
>
> While we acknowledge the importance of broader evaluations, collecting real-world data with reliable physical ground truth is non-trivial, often requiring specialized hardware and calibration. This challenge is especially pronounced for tasks involving **latent physical properties** (e.g., LOS, TDoA), which are not easily labeled.
>
> Nevertheless, we fully agree that expanding the real-world evaluation would significantly strengthen the paper. We will revise the manuscript to clarify our current positioning and will make every effort to include additional real-world experiments.
>
>
>
> ---
>
>
> **Regarding Active Sensing in Complex Scenarios**
>
> **Response:** We appreciate the reviewer's question about active sensing in cluttered environments. Our approach maintains strong performance in such scenarios for several reasons:
>
> Multiple reflectors create temporally separable echoes in the received signal. For range estimation, we focus on the first-arriving echo (nearest object). Thanks to low sound speed, with a 8 kHz chirp bandwidth, the reflections can be separated with a resolution of ~2.1 cm ($\frac{c}{2B}$​ [1]), allowing distinct identification of different reflecting objects.
>
> Our phase-aware encoder effectively captures temporal information, with phase information being particularly sensitive to small time shifts. This enables identification of the earliest echo even among multiple reflections from surfaces like windows, dashboards, and ceilings.
>
> Regarding Doppler effects, we observe minimal performance degradation under moderate motion. Chirp signals (i.e., FMCW) inherently resist Doppler shifts due to their linear time-frequency modulation. This is why it is used in radar systems for tracking fast-moving objects [1,2]. In our experiments, Doppler effects have negligible impact on range estimation accuracy.
>
> We will include these into our revised manuscript.
>
> **References:**
>
> [1] S. Rao. Introduction to mmWave Sensing: FMCW Radars. TI mmWave Training Series, 2017.
>
> [2] W. Mao, J. He, and L. Qiu. CAT: High-Precision Acoustic Motion Tracking. Proc. ACM MobiCom, 2016.
>
>
> ---
>
> **Regarding Impulse Response Comparison:**
>
> **Response:** Thanks. We will include visual comparisons between simulated and real CIR in the revision.
>
> **Preliminary examples of these comparisons** (LOS and NLOS) can be viewed at (We search for the most similar simulated CIR to the real case. Left: Real CIR, Right: Matched Simulated CIR):
>
> 1. https://imgur.com/aRQoP6H
> 2. https://imgur.com/xZ795Cb
> 3. https://imgur.com/XIYw2WX
> 4. https://imgur.com/5PfI8az
>
>
> ---
>
> **Regarding Related Work on Acoustic Localization or Radar work**
>
> **Response:** Thanks. We will dicuss them in the revison.

---

> > ### Comment · Reviewer_9yAo · 2025-04-03
> >
> > Thank you for addressing my concerns and including the additional experiments. After reading the rebuttals, I am now inclined to recommend Accept. Please ensure your new experiments are included in the revised version.

---

> > > ### Author Response · Authors · 2025-04-03
> > >
> > > Thank you for your thoughtful review and consideration. We're glad our clarifications and experiments addressed your concerns. These new experiments will be included in the revised version.

---

### Official Review · Reviewer_yVZE · 2025-03-13

**Overall Recommendation:** 4

**Summary:**

The paper proposes teaching Large Language Models (LLMs) to understand the physical world through sound. The authors created a physics-based simulator to generate a large audio dataset, AQA-PHY, annotated with physical phenomena like the Doppler effect and spatial relationships. They also developed a novel audio encoder that captures both the magnitude and phase of sound. By connecting this encoder to existing LLMs, they demonstrated the feasibility of enabling LLMs to achieve physical awareness in tasks such as detecting line-of-sight, estimating Doppler shifts, and localizing sound sources in both simulated and real-world environments.

**Claims And Evidence:**

1. The paper shows that it is possible to teach LLMs to reason about the physical world through sounds using experiments such as line of sight detection and doppler effect estimation.
2. Experiments are done on real-world scenarios to show the effectiveness of the proposed method in the real world.
3. Physical awareness can have a broad definition. The paper does not show tasks such as source separation or audio classification. Can the LLM understand the semantics of the sound?
4. There are no qualitative examples showing conversions between the LLM and end users.

**Essential References Not Discussed:**

Tang, Changli, et al. "Can Large Language Models Understand Spatial Audio?." Proc. Interspeech 2024. 2024.

**Experimental Designs Or Analyses:**

The experimental design is meaningful and captures real world scenarios.

**Methods And Evaluation Criteria:**

1. The experiments done on tasks such as doppler effect estimation and range estimation are meaningful to test if LLMs can reason about the physical world using audio input.
2. There are no experiments to show if LLM can understand the underlying semantics of the audio. Can it tell the difference between a car horn and a rooster crowing?

**Other Comments Or Suggestions:**

NA

**Other Strengths And Weaknesses:**

1. The paper proposed a novel audio encoder that processes magnitude and phase of audio signals outperforming Whisper.
2. The authors created a novel audio question-answering dataset to fine-tune LLM to teach them physical awareness.

**Questions For Authors:**

NA

**Relation To Broader Scientific Literature:**

I am not very familiar with this domain to be able to connect with the existing broader scientific literature. However, the impact statement does provide an insight where this work can plug into existing works. For instance, in embodied AI applications allowing robots to interact with various environments through sound.

**Theoretical Claims:**

NA

---

> ### Author Rebuttal · Authors · 2025-03-30
>
> # Response to Reviewer yVZE
>
> We sincerely thank the reviewer for the thoughtful and constructive feedback. We are encouraged by your recommendation! Our responses are as follows:
>
> ---
> **Regarding Semantics Understanding of Audio**
>
> **Response:** We agree that semantic understanding is an important dimension of audio perception. However, our current work focuses on teaching LLMs physical awareness—specifically, reasoning about phenomena such as Doppler shift, reverberation, line-of-sight, and time-of-flight. These capabilities are orthogonal and complementary to traditional audio tasks like classification or source separation, which have been extensively studied in existing models (e.g., Qwen-Audio, AudioGPT, Whisper, LTU, SpeechGPT).
>
> Our framework is fully compatible with semantic tasks. Since our simulator builds on AudioSet clips with known semantic labels, it can be naturally extended to support joint reasoning over both what a sound is (its semantic content) and how it propagates through space (the physical channel). We consider this a promising direction and will add it to the Limitations and Future Work section.
>
>
> ---
>
> **Regarding Absence of Qualitative Dialogue Examples**
>
> **Response:** Thanks for you suggestion. We would like to clarify that the primary scope of this paper is to validate the feasibility of enabling LLMs to understand physical phenomena through sound. Accordingly, we focus on single-turn QA tasks that directly test the model’s ability to reason about physical dynamics.
>
> Meawhile, we do preserve the natural language generation capabilities of the LLM through open-form QA, as demonstrated in Table 9. These examples show that the model can produce interpretable, step-by-step explanations of physical effects such as range estimation or LOS detection.
>
> Regarding multi-turn conversation, we agree this is a promising direction for future work. Notably, as shown in Appendix F, our prompt format already adopts a conversation-like structure, which we currently use in a single-turn setting. This format could be naturally extended to construct multi-turn dialogue datasets that allow LLMs to reason about physical phenomena over a series of interactions.
>
>
> ---
>
> **Regarding References Not Discussed**
>
> **Response:** Thank you for suggesting the reference "Can Large Language Models Understand Spatial Audio?" (Tang et al., 2024). We will include this reference and properly position our work relative to it.

---

> > ### Comment · Reviewer_yVZE · 2025-04-03
> >
> > I appreciate the authors for providing clarifications. I believe the rebuttal effectively addressed my questions. Additionally, I took into account the concerns of other reviewers and the authors’ response. To clarify, I am not super familiar with the field of acoustics or its related hardware. My knowledge is limited to text-to-audio models and other multimodal models that incorporate audio. From my perspective, the paper has made enough novel contributions. I will maintain my original rating of Accept unless other reviewers raise any major concerns.

---

### Official Review · Reviewer_3nj6 · 2025-03-14

**Overall Recommendation:** 3

**Summary:**

The paper introduces an approach to teach physical awareness to large language models (LLMs) through sound, using a physics-based audio simulator to create the AQA-PHY dataset. The dataset consists of 1 million audio-based question-answer pairs capturing phenomena such as Doppler effects, multipath, and spatial acoustics. It proposes a novel audio encoder that leverages both magnitude and phase information, outperforming baseline methods in tasks including LOS detection, Doppler estimation, and range estimation, with promising results in both simulated and real-world settings.

## Update after rebuttal

The rebuttal has addressed my major concerns. My rating has been increased.

**Claims And Evidence:**

The authors claimed that "extensive evaluation demonstrating state-of-the-art performance in both simulated and real-world environments." However, I have concerns about (1) very limited real-world data testing was conducted; (2) comparison to the Whisper encoder is not fair.

**Essential References Not Discussed:**

N/A

**Experimental Designs Or Analyses:**

The real-world experiments presented are interesting; however, they are quite limited due to the small dataset used and the evaluation conducted in only one specific vehicle audio environment.

**Methods And Evaluation Criteria:**

Regarding the method, it is unclear how spatial audio is modeled, as the current approach appears to use only mono audio signals. To effectively teach physical awareness, incorporating spatial audio would be crucial, as it allows models to learn spatial relationships from acoustic environments. However, this aspect is not clearly described in the paper.

**Other Comments Or Suggestions:**

N/A

**Other Strengths And Weaknesses:**

Overall, I think the authors explore an interesting research direction. However, I have several critical concerns. I would like summarize them here.

- Regarding the method, it is unclear to me if spatial audio is used in the audio encoder, as the current approach appears to use only mono audio signals. To effectively teach physical awareness, incorporating spatial audio would be crucial, as it allows models to learn spatial relationships from acoustic environments. However, this aspect is not clearly described in the paper.

- Whisper encoder is not a valid baseline. The authors build the dataset using sources from Audioset, which consists of ambient sounds, music, and speech. However, Whisper was trained only using speech data. I do not think it is a good baseline that can handle universal sound sources.

- The real-world experiments presented are interesting; however, they are quite limited due to the small dataset used and the evaluation conducted in only one specific vehicle audio environment. These experiments cannot validate if the trained model using the collected synthetic data can be effectively transferred to handle real-world sounds. The authors are suggested to collect a real-world testing dataset with more diverse sources.

**Questions For Authors:**

The authors are encouraged to address the raised major concerns.

**Relation To Broader Scientific Literature:**

This work extends existing multimodal LLMs like Llama3.1 and Qwen2 with physical awareness through sound.

**Theoretical Claims:**

N/A

---

> ### Author Rebuttal · Authors · 2025-03-30
>
> # Response to Reviewer 3nj6
>
>
> We thank the reviewer for the thoughtful and constructive feedback. We recognize that certain aspects of **our presentation may have led to confusion**, and we appreciate the opportunity to clarify the following key points:
> - Our method **does support spatial audio** using multi-channel input, both in simulation and real-world evaluation.
> - The use of **Whisper as a baseline** is justified and configured for a fair network structure comparison.
>
> Below are our responses.
>
> ---
> **Regarding Concern about Spatial Audio**
>
> **Response:** We respectfully clarify that **our method explicitly support spatial audio.**
>
> - Our system support both mono and multichannel audio input. For example, Direction-of-Arrival (DoA) estimation is one of our core tasks, which is inherently a fundamental spatial audio problem.
> - Tasks, such as DoA estimation, are trained and evaluated using two-channel input. For example, in Table 9, the questions for DoA Estimation explicitly include two `<Audio>` placeholders, corresponding to two audio channels from two microphones.
> - Our simulator (see Section 4.4) fully supports multi-microphone spatial audio simulation, including spatialized impulse responses and inter-channel phase differences.
> - Additionally, our real-world evaluation includes a left/right detection task using recordings from multiple microphones.
>
> We apologize for any lack of clarity in our presentation and will revise the manuscript to more explicitly highlight our support for multi-channel spatial audio.
>
>
> ---
>
> **Regarding Whisper as a Baseline**
>
> **Response:** We use Whisper as a structure-only baseline, not to compare pretrained capabilities. All encoders are fine-tuned on AQA-PHY to ensure a fair comparison focused purely on architectural differences. Importantly, Whisper is widely adopted as a standard audio encoder baseline.
>
> This setup allows for a fair comparison of encoder structure and their ability to adapt to physical reasoning tasks.
>
> We will clarify this setting in the revised manuscript to prevent misunderstanding.
>
> ---
>
> **Regarding Concerned about Real-World Evaluation**
>
> **Response:** We agree that the scope of our real-world evaluation is currently limited, and we sincerely appreciate the reviewer’s suggestion. Our primary goal in including the real-world experiment was to establish a **proof of concept**—demonstrating that models trained purely on synthetic data (AQA-PHY) can generalize effectively to real acoustic environments, thereby validating the feasibility of sim2real transfer.
>
> While we acknowledge the importance of broader evaluations, collecting real-world data with reliable physical ground truth is non-trivial, often requiring specialized hardware and calibration. This challenge is especially pronounced for tasks involving **latent physical properties** (e.g., LOS, TDoA), which are not easily labeled.
>
> Nevertheless, we fully agree that expanding the real-world evaluation would significantly strengthen the paper. We will revise the manuscript to clarify our current positioning and will make every effort to include additional real-world experiments.
>
>
> ---
>
> We appreciate the opportunity to clarify these points and hope our responses address the reviewer’s concerns. We hope these clarifications contribute to a clearer understanding of our work.

---

> > ### Comment · Reviewer_3nj6 · 2025-04-04
> >
> > Thanks for the response. The rebuttal has addressed my major concerns. My rating has been increased.

---

### Official Review · Reviewer_Gt9i · 2025-03-20

**Overall Recommendation:** 4

**Summary:**

The authors train an LLM to have knowledge of the physical world through acoustics. To do this, the authors create a large synthetic dataset of question-answer pairs that include audio from an acoustic simulator. The authors introduce an audio encoder that incorporates phase information, and they show that it improves performance on the acoustic tasks compared to an off-the-shelf encoder, that it works with multiple LLMs, and that it works with real world (non-simulated) data.

**Claims And Evidence:**

I believe that the primary claim, that training on this acoustic QA dataset teaches the model physical awareness, is not entirely supported. It is true that the model learns to compute various acoustic quantities from raw audio. But I feel that for the model to be "aware" and "understand physical phenomena" -- rather than simply be an acoustic calculator -- it should be able to use acoustics to solve other, useful tasks. The authors suggest several possible tasks in Table 3 (Appendix A).

(I understand this usage of "aware" and "understand" may be contentious, but I believe this interpretation is within the scope of the paper.)

**Essential References Not Discussed:**

There should be some discussion about how this work relates to other acoustic/physical simulators like SoundSpaces (Chen at al. 2022), as well as similar datasets SpatialSoundQA (Zheng et al. 2024).

Chen et al., 2022: [SoundSpaces 2.0: A Simulation Platform for Visual-Acoustic Learning](https://arxiv.org/abs/2206.08312)

**Experimental Designs Or Analyses:**

The experiments (sole/merged testing, ablations, and real world) seem sound.

**Methods And Evaluation Criteria:**

The setup of the acoustic simulation is simple and modular. One (relatively minor) missing detail is the setup of the 32-layer Transformer in the audio encoder. It is stated that it is a 32-layer Transformer, but does everything match the Whisper Large encoder?

I find a few issues with the evaluation:
1. The accuracy of the intermediate steps is never evaluated.
2. The dataset essentially teaches the LLM to be a calculator for acoustics, rather than a tool that learns to apply acoustics to solve another task (see "Claims" above). I believe the real-world dataset may help here. e.g. does the LLM know that a sound is coming from "inside the car" instead of simply "out of LOS"? (This example is from Figure 1a.) If so, that could indicate the LLM is able to use its "awareness" to answer questions.
3. In Section 6.4, it is not clear what data the model is trained on. (i.e., is it trained on any real data, or only simulated?)
4. There is no quantitative comparison against the BAT model and its Spatial-AST encoder (Zheng et al., 2024) - this could done be on AQA-PHY or on SpatialsoundQA.

**Other Comments Or Suggestions:**

1. Two typos:

    a. Line 290, "finial"

    b. In Section 6.2 Line 352, "5 percentages improvement" should be more like "5 percentage points of improvement".

**Other Strengths And Weaknesses:**

Other strengths:
1. The simulated-to-real experiment shows that knowledge learned form the AQA-PHY dataset can transfer well to real data.

**Questions For Authors:**

1. The primary difference between the proposed encoder and Whisper is the addition of phase information. Given that, why does Whisper perform so much worse on LOS detection, even when it is the sole task? (Table 2)

2. How was the LoRA rank determined (Table 8)? With higher ranks, is there a trade-off in what the audio encoder learns versus the LLM?

3. Examining the example QA pairs in Table 9, some of the Close & Open question forms seem very similar, like for LOS Detection and Multipath analysis. Does the model not become confused about which form to output? And if they are so similar, what benefit does training on the Closed form give here?

4. How is the LM trained to do function calls for Range Estimation? In Table 9 I do not see an example of this (and more generally I don't see examples of the "active sensing" that this task is supposed to give).

5. How is the "final answer" extracted from the LM response, so that it can be scored with BCA, MAE, etc.?

**Relation To Broader Scientific Literature:**

The authors show that training a phase-aware audio encoder can allow LLMs to reliably compute acoustic quantities through natural language. Unlike related spatial audio processing work (BAT, Zheng et al. (2024), the proposed method does not require preprocessing all audio inputs to calculate their relative phase.

**Theoretical Claims:**

I checked the formulation of the channel impulse response (1) - (8).

---

> ### Author Rebuttal · Authors · 2025-03-31
>
> # Response to Reviewer Gt9i
> We sincerely thank the reviewer for the thoughtful and constructive feedback. We are encouraged by your recommendation! Our responses are as follows:
>
> ---
> **Regarding Physcial Awareness vs. Acoustic Calculator**
>
> **Response:**
> Thank you for raising this insightful question!
>
> We believe physical awareness emerges progressively as model scale, task diversity, and reasoning complexity increase—similar to trends in LLMs.
> Thus, we (1) use a physics-based simulator to generate diverse, controllable acoustic scenes, and (2) design open-form QA tasks that encourage causal reasoning, not just output a number or a label.
>
> Rather than treating the 'acoustic calculator' as an end goal, we view it as a **probing tool to assess whether LLMs can interpret raw waveforms**.
>
> For open-ended QAs, the goal is for the LLM to understand the underlying physical phenomenon—not merely to compute a value, but to reason about causes and express that understanding in natural language.
>
> We will clarify this.
>
> ---
> **Regarding Intermediate Step Accuracy**
>
> **Response:** We do report intermediate prediction errors in Table 2: DoA and range estimation results are presented in both time-domain estimation and final output (see Table footnote).
>
> Further, we include two figures showing how TDoA error correlates with DoA error, and how ToF error maps to distance error. In the range plot, the discrete vertical steps are due to rounding—our dataset uses integer-valued distances (in m), so small variations in ToF can lead to quantization in distance error. These results show that the model extracts and reasons over physical intermediate variables.
>
> **Results:**
> - DoA: https://imgur.com/VkZlSvz
> - Range: https://imgur.com/0qBVk9K
>
> ---
> **Regarding Whisper underperforms on LOS Detection**
>
> **Response:**  Magnitude features capture signal content—e.g., what is being said.  One good example is AudioMAE, which can generate audio from magnitude only. In contrast, **phase encodes fine-grained temporal cues**, revealing how a signal propagates through space.
>
> LOS detection relies on phase cues:
> - A LOS signal shows a sharp onset and consistent phase.
> - Reflected paths exhibit delayed or disrupted phase due to multipath.
>
> **These distinctions are often invisible in magnitude but clear in phase**, making phase essential. We will explain this in the revision.
>
> ---
> **Regarding Training Data**
>
> **Response:** The model is trained only on simulated data. Real-world results are zero-shot evaluations.
>
> ---
> **Regarding Comparison with BAT**
>
> **Response:** While BAT is a pioneering work on spatial audio reasoning, our work targets physics-based understanding, with different goals, tasks, and evaluation formats (see response to reviewer t8Co), making direct comparison nontrivial.
>
> The only overlapping task is left/right direction classification. We conducted a quick evaluation of BAT model on AQA-PHY and observed 84.8% accuracy, compared to 99.4% achieved by our encoder + Qwen2. We note BAT’s performance may be affected by setup differences (e.g., sampling rate, training dataset).
>
> We will clarify these.
>
> ---
> **Regarding Audio Encoder Architecture**
>
> **Response:**  Our encoder largely follows Whisper-large for a fair comparison, with the main difference being the input: we use [magnitude, phase_sin, phase_cos] and corresponding convolution layers.
>
> ---
> **Regarding LoRA Rank**
>
> **Response:** We kept the LoRA rank relatively low (=8) to preserve the LLM’s language ability, as most representation learning occurs in the audio encoder. This choice is aligned with the model-agnostic property of our model.
> Higher ranks showed minimal benefit but increased cost (see attached table). Due to time limits, this is a preliminary study. We will conduct a more detailed one.
>
> **Initial Study of LoRa Rank:** https://imgur.com/p5HfJMQ
>
> ---
> **Regarding Similar Prompts**
>
> **Response:** Thanks for pointing this out. While Table 9 examples seem similar, our open-form prompts are more diverse. Examples include:
> - "Could you evaluate the effect of multipath interactions within the given audio piece?"
> - "What is the extent to which the multipath effect manifests in the sound?"
>
> For training, LLMs learns to distinguish based on prompt: closed-form prompts yield fixed answers, while open-form ones encourage open-form answers.
>
> We will revise these.
>
> ---
> **Regarding Function Call**
>
> **Response:** Sorry for the confusion. We do not train the LLM to perform function calls. Rather, we envision active sensing can be triggered via function calls, like activating a speaker to emit pulses. we focus on showing that the LLM can process the resulting echo to estimate range. We will clarify this.
>
> ---
> **Regarding Final Answer Extraction**
>
> **Response:** We use GPT-4o to extract the final answer.
>
> ---
> **Regarding Related Simulator and SpatialSoundQA**
>
> **Response:** Thanks. We will discuss them.
>
> ---
> **Regarding Typos Correction**
>
> **Response**: Thanks. We will fix them.

---

### Official Review · Reviewer_t8Co · 2025-03-22

**Overall Recommendation:** 3

**Summary:**

The paper presents a novel method to imbue large language models with physical awareness through sound by using a physics-based channel simulator that synthesizes realistic acoustic data, simulating phenomena such as the Doppler effect, multipath reflections, and LOS conditions. The authors design an audio encoder that processes both the magnitude and phase (via sine and cosine components) of the audio signal, enabling the model to capture subtle physical characteristics. This encoder is integrated with LLMs and trained on the AQA-PHY dataset—a large-scale collection of one million <Audio, Question, Answer> tuples—using supervised fine-tuning. Extensive evaluations across passive sensing tasks and active sensing tasks demonstrate the method’s effectiveness and model-agnostic improvements.

**Claims And Evidence:**

Yes. Evaluations across tasks like LOS detection, Doppler estimation, DoA estimation, multipath analysis, and range estimation demonstrate claims in the paper.

**Essential References Not Discussed:**

N/A

**Experimental Designs Or Analyses:**

All looks good.

**Methods And Evaluation Criteria:**

Yes.
(1) LOS Detection: binary classification accuracy (BCA, ↑);
(2) Doppler estimation, Mean Absolute Error (MAE);
(3) DoA estimation: MAE;
(4) Multipath Analysis: Triple-class Classification Accuracy (TCA, ↑);
(5) Range Estimation: Relative Error Percentage (REP, ↓).

All evaluation criteria make sense.
For proposed method, which incorporates phase into its encoder to better capture physical phenomena.

**Other Comments Or Suggestions:**

Line 045 should have a period.

**Other Strengths And Weaknesses:**

Strengths:
1. The paper is easy to follow.
2. For the first time, this paper established a connection between physical acoustic simulation techniques and LLMs, which is quite novel.
3. Sim2Real evaluation is valuable.

Weaknesses:
1. Compared to previous work—BAT, this paper is more likely to explain things from a physical perspective, but essentially it does not seem to offer sufficient innovation. Many aspects, such as the model architecture and encoder design, are fundamentally inherited from the settings in BAT.
2. In sim2real case, the model performs well on LOS and Left/Right detection. However, since these tasks are basically binaural classification questions, which makes them quite easier to answer. I wonder if it's possible for authors to also evaluate more complicated tasks to show the model really generalize well on real situation.

[1] BAT: Learning to reason about spatial sounds with large language models. Zheng, Z., Peng, P., Ma, Z., Chen, X., Choi, E., and Harwath, D.

**Questions For Authors:**

Add a random performance baseline in Tab 2 makes it easier to understand the overall performance relative to chance.

**Relation To Broader Scientific Literature:**

The paper extends established signal processing and acoustic simulation techniques—such as modeling the Doppler effect and multipath reflections—to enhance LLMs with physical awareness, building on prior work in audio understanding (e.g., BAT).

**Theoretical Claims:**

The theoretical claims for LOS ad Early Reflections, Reverberation, Doppler Effect, Microphone Array are correct.

---

> ### Author Rebuttal · Authors · 2025-03-30
>
> # Response to Reviewer t8Co
>
> We sincerely thank the reviewer for the thoughtful and constructive feedback. We are encouraged by your recommendation! Our responses are as follows:
>
>  ---
> **Regarding Innovation Compared to BAT**
>
> **Response:** We greatly appreciate BAT's pioneering work in bringing spatial audio understanding to LLMs, which partially inspires our work.  However, our work addresses fundamentally different problems and technical challenges, requiring distinct approaches:
> - **Different goals and focus:** BAT primarily addresses spatial audio perception for auditory scene understanding (e.g., "Is sound A to the left of sound B?"), while our work focuses on teaching LLMs to understand fundamental physical phenomena through sound. We enable LLMs to comprehend physics principles rather than just spatial relationships.
> - **Broader range of physical awareness tasks:** We explore physical tasks entirely absent from BAT, including Doppler effect estimation, multipath analysis, and active sensing capabilities (range estimation). Understanding dynamic phenomena like the Doppler effect requires fundamentally different modeling than BAT's spatial tasks.
> - **More scalable phase-aware audio encoding:** Unlike BAT, which uses Interaural Phase Difference (IPD) between microphone pairs, our encoder extracts phase information directly from individual channels. In this way, we preserve phase information **without introducing quadratic growth** in computations as microphone count increases. Notably, our encoder works effectively even with single-channel audio (e..g, LOS detection, Multipath Analysis), while BAT's approach reduces to a magnitude-only encoder in such cases.
> - **Open-format QA evaluation:** We introduce open-format QA to assess deeper physical understanding. While BAT primarily uses LLMs for classification or regression with fixed outputs, our approach leverages LLMs' reasoning capabilities through natural language explanations. This allows us to evaluate whether models truly understand physical phenomena rather than simply mapping inputs to pre-defined labels.
> - **Comprehensive physics-based channel simulator:** We've developed a dedicated simulator that models acoustic physical phenomena (LOS paths, early reflections, reverberation, Doppler effects) in controlled ways. This enables precise generation of diverse training data with accurate physical annotations, going beyond BAT's focus on spatial localization in static environments.
>
> ---
>
> **Regarding Real-World Evaluation**
>
> **Response:** We agree that the scope of our real-world evaluation is currently limited, and we sincerely appreciate the reviewer’s suggestion. Our primary goal in including the real-world experiment was to establish a **proof of concept**—demonstrating that models trained purely on synthetic data (AQA-PHY) can generalize effectively to real acoustic environments, thereby validating the feasibility of sim2real transfer.
>
> While we acknowledge the importance of broader evaluations, collecting real-world data with reliable physical ground truth is non-trivial, often requiring specialized hardware and calibration. This challenge is especially pronounced for tasks involving **latent physical properties** (e.g., LOS, TDoA), which are not easily labeled.
>
> Nevertheless, we fully agree that expanding the real-world evaluation would significantly strengthen the paper. We will revise the manuscript to clarify our current positioning and will make every effort to include additional real-world experiments.
>
> ---
>
> **Regarding Random Performance Baseline**
>
> **Response:** Thank you for this good suggestion! We will include random performance baselines in Table 2 for the final version. For classification tasks (BCA and TCA), the random baselines are straightforward. For regression tasks, we calculate the expected error when randomly sampling outputs within the possible value range:
> - LOS Detection ($BCA$): 0.5
> - Doppler Estimation ($MAE_f$): 10.0 (assuming random outputs within the possible frequency shift range of [-15%, 15%], with maximum speed of 50 m/s and sound speed of 343 m/s)
> - DoA Estimation ($MAE_t$): 66.7 (assuming random outputs within the possible TDoA range of [-100, 100])
> - Multipath Analysis ($TCA$): 0.33
> - Range Estimation ($REP$): 33.3 (assuming random outputs yielding relative errors between 0-100%)
>
> We believe these baselines will highlight the improvements our approach achieves. Thank you again.
>
> ---
>
> **Regarding Typo Correction**
>
> **Response:** Thanks. We will fix this in the revision.

---

### Decision · Program_Chairs · 2025-05-01

**Decision:**

Accept (poster)

**Comment:**

This paper proposes a novel approach to teaching large language models (LLMs) to understand the physical world through sound. The authors introduce AQA-PHY, a large-scale audio dataset generated using a physics-based simulator and annotated with physical phenomena such as the Doppler effect and line-of-sight relationships. They also develop a new audio encoder that captures both magnitude and phase information, which is then connected to LLMs to enable physical reasoning tasks like Doppler shift estimation, sound source localization, and line-of-sight detection in both simulated and real-world settings. The authors addressed reviewers’ concerns thoroughly in their rebuttal, providing additional experiments and clarifications. All reviewers expressed satisfaction with the response and recommended acceptance.